# Hierarchical Cautious Optimization for Semi-Supervised Medical Image Segmentation with Limited Labeled Data

## Abstract

Semi-supervised learning (SSL) effectively addresses limited labeled data challenges in volumetric medical image segmentation by leveraging both ground-truth labels and pseudo-labels from unlabeled data. However, conventional optimizers treat gradients from labeled and unlabeled sources equally, often leading to either over-trust or over-rejection of pseudo-label signals. We introduce Hierarchical Cautious Optimization (HCO), which establishes a trust hierarchy between gradient sources. HCO computes momentum estimates using only gradients from labeled data and incorporates unlabeled gradients only when they align with this trusted direction. Our approach integrates into existing momentum-based optimizers with minimal implementation effort and computational cost. Evaluations across three datasets demonstrate consistent performance improvements, particularly on a challenging fetal MRI dataset where Dice scores for fetal lungs and liver increased from 0.68 to 0.84 and 0.71 to 0.82, respectively. The consistent gains across optimizers and datasets, combined with minimal implementation overhead, position HCO as a practical enhancement for existing SSL medical segmentation pipelines.

## 1 Introduction

Deep convolutional neural networks (DCNNs) have proven their effectiveness for volumetric medical image segmentation. However, they require large, expert-annotated datasets that are often unavailable and difficult to create (Cheplygina et al., 2019; Litjens et al., 2017). Semi-supervised learning (SSL) methods aim to reduce the annotation burden by using a few annotated and many unannotated samples (Ouali et al., 2020). Predominant SSL strategies involve generating supervisory signals from unlabeled data, often via pseudo-labeling or enforcing consistency across augmentations (Sohn et al., 2020; Tarvainen & Valpola, 2017). Although recent research has focused on improving pseudo-labels (Bai et al., 2023; Huang et al., 2023; Luo et al., 2021; Shen et al., 2023; Song & Wang, 2024; Wu et al., 2021; Yu et al., 2019; Zhou et al., 2023), the optimization process has received much less attention.

SSL methods typically minimize a combined loss function, $\mathcal{L} = \mathcal{L}_{\mathrm{L}} + \mathcal{L}_{\mathrm{U}}$, where $\mathcal{L}_{\mathrm{L}}$ and $\mathcal{L}_{\mathrm{U}}$ are the losses over labeled and unlabeled data respectively. Crucially, conventional optimizers like SGD, Adam, or AdamW (Kingma & Ba, 2015; Loshchilov & Hutter, 2019) are agnostic to the source of the gradient. They treat the precise, ground-truth supervision from $\nabla\mathcal{L}_L$ as equivalent to the noisy, potentially erratic signals from $\nabla\mathcal{L}_U$. This symmetry is dangerous: unlabeled gradients are prone to high variance and noise (Arazo et al., 2020). Treating them equally exposes the model to confirmation bias (reinforcing incorrect pseudo-labels) and co-training collapse (synchronizing errors across views) (Chen et al., 2021). With pseudo-labeling and consistency techniques reaching maturity, revisiting optimization methods offers a promising avenue to further unlock SSL potential.

We propose Hierarchical Cautious Optimization (HCO), a framework for momentum-based optimizers that establishes a trust hierarchy between labeled and unlabeled gradients. HCO prioritizes reliable supervision signals by computing momentum estimates solely from labeled gradients $\nabla\mathcal{L}_{\mathrm{L}}$, establishing a trusted optimization direction. HCO incorporates unlabeled gradients $\nabla\mathcal{L}_{\mathrm{U}}$ cautiously: they update momentum only when aligned with the trusted labeled momentum. This

ensures unlabeled gradients amplify but never contradict the labeled direction. Our hierarchical design directly addresses confirmation bias and co-training collapse by preventing conflicting gradients from corrupting the optimization trajectory.

We validate HCO across three diverse volumetric benchmarks: Left Atrium (Xiong et al., 2021), Pancreas-CT (Roth et al., 2015), and a challenging private fetal MRI dataset. In all settings, HCO functions as a drop-in replacement for standard optimizers, yielding consistent accuracy gains without architectural changes.

Our contributions are fourfold: (1) **HCO**, a novel optimization framework that establishes a trust hierarchy between labeled and unlabeled gradients; (2) **Convergence Analysis**, proving that HCO preserves convergence guarantees of base optimizers under standard assumptions (Appendix A); (3) **Mechanistic Understanding**, showing through supervised-unsupervised loss dynamics that HCO promotes sustained utilization of unlabeled signals (Section 4.5); and (4) **Empirical and Practical Validation**, demonstrating consistent performance gains across three datasets with minimal code changes and negligible computational overhead.

## 2 RELATED WORK

**Semi-Supervised Learning for Segmentation.** Semi-Supervised Learning (SSL) for segmentation leverages limited labeled data ($D_L$) alongside abundant unlabeled data ($D_U$). Core techniques include pseudo-labeling (Lee, 2013) and consistency regularization for stable pseudo-labels under perturbations (Sajjadi et al., 2016; Laine & Aila, 2017). The Mean Teacher (MT) framework (Tarvainen & Valpola, 2017) uses a teacher network updated as an exponential moving average (EMA) of the student weights, providing temporally smoothed predictions for guidance. Many variants adapt MT to improve pseudo-label quality: UA-MT estimates epistemic uncertainty with Monte Carlo dropout and discards unreliable targets (Yu et al., 2019); DTC and SASSNet add level-set regression and dual-task consistency to impose geometric constraints on unlabeled cases (Luo et al., 2021; Li et al., 2020); other works refine pseudo-labels through uncertainty rectification (Zheng & Yang, 2021), and exploit strong spatial augmentations such as CutMix and Bidirectional Copy-Paste (BCP) (Yun et al., 2019; Bai et al., 2023). Multi-teacher extensions diversify supervision instead of relying on a single EMA teacher: CMT couples two networks with Cross Pseudo Supervision (Shen et al., 2023); AD-MT alternates updates of two EMA teachers and resolves their disagreements with a conflict-aware ensembling rule (Zhao et al., 2024); PMT maintains temporally staggered teacher states and filters pseudo-labels based on temporal consistency and confidence (Gao et al., 2024).

**Robust Optimization Strategies.** Previous work addresses noisy or conflicting gradients using sign-based rules (Bernstein et al., 2018; Chen et al., 2023), clipping/thresholding, and per-parameter gating as in Cautious Optimizers (CO) (Liang et al., 2024), which blocks steps misaligned with running momentum. These methods are *objective-symmetric*: they suppress misaligned components regardless of the source. In SSL, reliability is *asymmetric*. Labeled gradients are trusted; unlabeled gradients require verification. HCO builds this asymmetry into the optimizer: it forms momentum solely from $\nabla \mathcal{L}_L$, then filters $\nabla \mathcal{L}_U$ element-wise by sign agreement with that trusted direction. Aligned components may refine the state, but cannot redefine it. This prevents unlabeled components from contaminating future updates through moment buffers, addressing a failure mode that CO does not handle.

## 3 METHOD

Hierarchical Cautious Optimization (HCO) augments any momentum-based optimizer with an explicit trust hierarchy that distinguishes gradients computed from labeled data from those obtained on unlabeled data (Fig. 1). We enforce the hierarchy in two steps: (i) we first update the optimizer state with the reliable labeled gradient, establishing a trusted update direction; (ii) we allow only the components of the unlabeled gradient that align with that trusted direction to refine the momentum buffer and influence the parameter update. This section describes the resulting algorithm, introduces notation, and details the role of variables appearing in Algorithms 1–2.

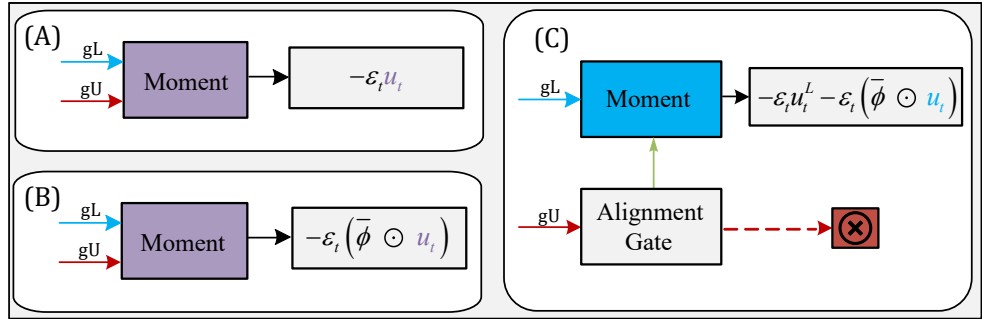

Figure 1: **Comparison of Optimization Mechanisms.** **(A)** Standard momentum optimizer integrates labeled ($g_L$) and unlabeled ($g_U$) gradients symmetrically. **(B)** Cautious Optimization (CO) retains symmetric accumulation but filters updates elementwise, allowing steps only when the gradient sign agrees with momentum. **(C)** HCO establishes a labeled-driven momentum and uses an alignment gate to admit only unlabeled components consistent with this direction (green), excluding conflicting components (dashed red). This enforces a source-aware trust hierarchy. Notation follows in the glossary below.

## 3.1 Notation and Variable Glossary

At iteration $t$ the network parameters are $w_t \in \mathbb{R}^d$. Gradients and moving averages are of the same dimensionality. Unless stated otherwise all operations are element-wise.

$w_t$ Network parameters after $t$ updates.

$g_t^L, g_t^U$ Gradients computed from labeled and unlabeled data, respectively.

$m_t, v_t$ Exponential moving average buffers of the first and second moments.

$u_t^L$ Trusted momentum direction derived solely from labeled gradients $g_t^L$.

$u_t$ Final cautious momentum after selectively incorporating aligned components of $g_t^U$.

$\mathbb{I}(\cdot)$ Element-wise indicator function returning 1 for true conditions and 0 for false ones.

$\phi_t$ Binary alignment masks that identify components of $g_t^U$ aligned with $u_t^L$ and $u_t$.

$\overline{\phi}_t$ Scaled binary masks that preserve update magnitude despite selective component filtering.

**Hyperparameters:** $\varepsilon_t$ (learning rate), $\beta_{1,2}$ (decay factors), $\epsilon$ (numerical stability constant).

## 3.2 Hierarchical Cautious AdamW

Algorithm 1 presents our Hierarchical Cautious AdamW optimizer. The algorithm proceeds through three distinct phases:

**Step 1. Labeled step (lines 4-10).** The labeled gradient $g_t^L$ is accumulated into the moment buffers, yielding the bias-corrected direction $u_t^L = \hat{m}_t/(\sqrt{\hat{v}_t} + \epsilon)$. Since it is informed solely by ground-truth labels, $u_t^L$ defines a direction that is *trusted* by construction.

**Step 2. Unlabeled Momentum Refinement (UMR) step (lines 11-17).** The unlabeled gradient $g_t^U$ is masked by $\phi_t = \mathbb{I}(u_t^L \odot g_t^U > 0)$ to retain only components aligned with the labeled direction. The filtered gradient $g_t^U \odot \phi_t$ updates the moment buffers, which are then bias-corrected following standard Adam to yield the refined direction $u_t$.

**Algorithm 1** Hierarchical Cautious AdamW

**Require:** Initial parameters $w_0$, step sizes $\{\varepsilon_t\}$, $\beta_1, \beta_2 \in [0, 1)$, weight decay $\gamma \geq 0$, stability constant $\epsilon > 0$.
1: Initialize $t \leftarrow 0, m_0 \leftarrow 0, v_0 \leftarrow 0$
2: **while** not converged **do**
3:    $t \leftarrow t + 1$
4:    *// Labeled step (Compute trusted direction)*
5:    Compute $g_t^L \leftarrow \nabla_w \mathcal{L}_{\text{labeled}}(w_{t-1})$
6:    $m_t \leftarrow \beta_1 m_{t-1} + (1 - \beta_1)g_t^L$
7:    $v_t \leftarrow \beta_2 v_{t-1} + (1 - \beta_2)(g_t^L)^2$
8:    $\hat{m}_t \leftarrow m_t/(1 - \beta_1^t)$
9:    $\hat{v}_t \leftarrow v_t/(1 - \beta_2^t)$
10:   $u_t^L \leftarrow \hat{m}_t/(\sqrt{\hat{v}_t} + \epsilon)$
11:   Compute $g_t^U \leftarrow \nabla_w \mathcal{L}_{\text{unlabeled}}(w_{t-1})$
12:   $\phi_t \leftarrow \mathbb{I}(u_t^L \odot g_t^U > 0)$
13:   $m_t \leftarrow m_t + (1 - \beta_1)(g_t^U \odot \phi_t)$
14:   $v_t \leftarrow v_t + (1 - \beta_2)(g_t^U \odot \phi_t)^2$
15:   $\hat{m}_t \leftarrow m_t/(1 - \beta_1^t)$
16:   $\hat{v}_t \leftarrow v_t/(1 - \beta_2^t)$
17:   $u_t \leftarrow \hat{m}_t/(\sqrt{\hat{v}_t} + \epsilon)$
18:   $\overline{\phi}_t \leftarrow \phi_t/\max(\text{mean}(\phi_t), \epsilon)$
19:   $w_t \leftarrow w_{t-1} - \varepsilon_t u_t^L - \varepsilon_t(\overline{\phi}_t \odot u_t)$
20:   $w_t \leftarrow w_t - \varepsilon_t \gamma w_t$
21: **end while**
22: **return** $w_t$

**Algorithm 2** Hierarchical Cautious SGD

**Require:** Initial parameters $w_0$, step sizes $\{\varepsilon_t\}$, momentum decay $\beta_1 \in [0, 1)$, weight decay $\gamma \geq 0$, stability constant $\epsilon > 0$.
1: Initialize $t \leftarrow 0, m_0 \leftarrow 0$
2: **while** not converged **do**
3:    $t \leftarrow t + 1$
4:    *// Labeled step (Compute labeled momentum)*
5:    Compute $g_t^L \leftarrow \nabla_w \mathcal{L}_{\text{labeled}}(w_{t-1})$
6:    $m_t \leftarrow \beta_1 m_{t-1} + g_t^L$
7:    $m_t^L \leftarrow m_t$ *// Store momentum after Labeled step*
8:    *// Unlabeled step (Update momentum)*
9:    Compute $g_t^U \leftarrow \nabla_w \mathcal{L}_{\text{unlabeled}}(w_{t-1})$
10:   *// Align $g_t^U$ with labeled momentum $m_t^L$*
11:   $\phi_t \leftarrow \mathbb{I}(m_t^L \odot g_t^U > 0)$
12:   $\overline{\phi}_t \leftarrow \phi_t/\max(\text{mean}(\phi_t), \epsilon)$
13:   *// Update momentum with aligned $g_t^U$*
14:   $m_t \leftarrow \beta_1 m_t + (g_t^U \odot \overline{\phi}_t)$
15:   $w_t \leftarrow w_{t-1} - \varepsilon_t m_t^L - \varepsilon_t(m_t \odot \overline{\phi}_t)$
16:   *// Apply weight decay*
17:   $w_t \leftarrow w_t - \varepsilon_t \gamma w_t$
18: **end while**
19: **return** $w_t$

**Step 3. Cautious gating and parameter update (lines 18-20).** To maintain a consistent update magnitude when masking, we normalize the binary gate by its clipped mean, yielding $\overline{\phi}_t$. This produces a rescaled gating mask that preserves magnitude while filtering out conflicting components. The final weight change is $\Delta w_t = -\varepsilon_t u_t^L - \varepsilon_t(\overline{\phi}_t \odot u_t)$, combining the trusted labeled direction with the UMR-refined direction modulated by the normalized gate. This ensures the unlabeled adjustment only reinforces components already endorsed by the labeled gradient. Standard decoupled weight decay completes the update process.

### 3.3 HIERARCHICAL CAUTIOUS SGD

Algorithm 2 applies the same hierarchical gating and masking logic to SGD, using a single velocity buffer instead of Adam moments. This introduces essentially no overhead beyond one additional element-wise mask and a dot-product per iteration.

### 3.4 CONVERGENCE GUARANTEE

Under standard assumptions of $L$-smoothness and bounded stochastic gradient variance (Reddi et al., 2019), we prove in Appendix A that AdamW$_{\text{HCO}}$ maintains the same $\mathcal{O}(1/\sqrt{T})$ convergence rate to first-order stationary points as AdamW, where $T$ is the total number of optimization steps. Our hierarchical masking framework therefore preserves the theoretical guarantees of the base optimizer.

### 3.5 COMPUTATIONAL COST

Since alignment checks are element-wise and reuse already available quantities ($u_t^L$, $u_t$, $g_t^U$), the extra wall-clock overhead is below $1\%$ in our PyTorch implementation (Supplementary, 4).

## 4 EXPERIMENTS

**Datasets:** We use the following three datasets:

**1. Left-Atrium MRI.** The Left Atrial Segmentation Challenge public dataset (Xiong et al., 2021) consisting of 100 3D GE-MRI scans with segmentation labels. Pre-processing and data splits follow standard protocols from prior SSL segmentation work (Wu et al., 2022; Bai et al., 2023).

**2. Pancreas-CT.** The Pancreas NIH public dataset (Roth et al., 2015) consisting of 82 abdominal contrast-enhanced CT scans with manual pancreas contours delineations. Pre-processing as in (Shi et al., 2022).

**3. Fetal-MRI.** In-house fetal body MRI dataset of 92 manually-labeled and 600 unlabeled scans (gestational age 28–39 weeks; 3T Siemens True Fast Imaging with Steady-State Precession (TRUFI) sequence acquisitions; 0.78×0.78×2 mm voxels). All scans were retrospectively collected and de-identified under IRB approval. We held out 50 labeled cases for test and 2 for validation. (Full acquisition, annotation, and inter-observer details in Appendix B.)

**Evaluation Metrics**: We report four standard segmentation metrics: Dice (%) and Jaccard (%) scores for the segmentation region, and Average Surface Distance (ASD) and 95% Hausdorff Distance (HD95) for the segmentation contour.

**Implementation** All experiments were implemented in PyTorch and run on NVIDIA GeForce RTX 4090 GPUs with fixed random seeds for reproducibility. Unless otherwise specified, all hyper-parameters for each framework follow those of the original implementations. The total compute for the reported experiments was approximately 300 GPU hours, and no significant additional compute was used for preliminary explorations or hyperparameter tuning beyond this.

**Studies** We evaluate our method with five studies as follows:

- **Study 1: BCP on the Left-Atrium-MRI dataset.** Using BCP (Bai et al., 2023), a strong SSL baseline, we assess $SGD_{HCO}$ on the Left Atrial dataset (Xiong et al., 2021). We quantify its gains in Dice and surface metrics over conventional SGD.

- **Study 2: BCP on the Pancreas-CT dataset** Using BCP (Bai et al., 2023), we apply $Adam_{HCO}$ to the NIH Pancreas-CT dataset (Roth et al., 2015), thereby changing optimizer, imaging modality, and anatomical target. We quantify its gains in Dice, Jaccard, and surface metrics over standard Adam.

- **Study 3: CMT on the Fetal-MRI dataset.** Within the CMT consistency framework (Shen et al., 2023), we apply $AdamW_{HCO}$ to the fetal MRI dataset, thereby changing SSL framework, optimizer, and anatomical target. We report Dice and surface gains over AdamW.

- **Study 4: Optimizer Ablations.** On the fetal MRI task, we ablate hierarchy. We measure each variant's effect via changes in Dice and surface distance metrics versus AdamW.

- **Study 5: Impact of HCO on Supervised–Unsupervised Loss Dynamics.** For both BCP and CMT setups, we monitor $\mathcal{L}_{unlabeled}/\mathcal{L}_{labeled}$ over training to quantify how much the HCO variants preserve the unlabeled signal.

In all studies, the pre-processing and training protocols followed the exact same host framework. The only change was the optimizer, SGD, Adam, AdamW, to their HCO variants. For each study we indicate the number of independent runs and the statistical test employed.

### 4.1 STUDY 1: BCP ON THE LEFT-ATRIUM-MRI DATASET

**Objective.** Evaluate the performance improvement of HCO when replacing standard SGD in the BCP framework.

**Dataset and protocol.** We follow the BCP setup (Bai et al., 2023) with the 3D V-Net backbone (Milletari et al., 2016). Datasets were resampled to 1.25 mm isotropic and randomly cropped to $112 \times 112 \times 80$ patches, with on-the-fly rotations and flips. Training consisted of a pre-training phase (2K iterations) and a self-training phase (15K iterations) using SGD with an initial learning rate of

Table 1: Left-Atrium MRI segmentation results (20 test cases; mean (std) over 2 seeds). **VNet** and **BCP** results for SGD, $SGD_{CO}$, and $SGD_{HCO}$. Metrics: Dice/Jaccard (%), ASD (voxels), HD95 (voxels). Gray rows: BCP results from (Bai et al., 2023); (Repr): our reproduction. Bold indicates the best result per group and metric.

| Method | Optimizer | Scans Used L/U | Metrics | | | |
|---|---|---|---|---|---|---|
| | | | Dice ↑ | Jaccard ↑ | ASD ↓ | HD95 ↓ |
| *V-Net Baselines (Supervised)* | | | | | | |
| V-Net | SGD | 4/0 | 52.55 | 39.60 | 4.91 | 47.05 |
| V-Net | SGD | 8/0 | 82.74 | 71.72 | 3.26 | 13.35 |
| V-Net | SGD | 80/0 | 91.47 | 84.36 | 1.51 | 5.18 |
| *Semi-Supervised with BCP (4 Labeled)* | | | | | | |
| BCP (Paper) | SGD | 4/76 | 88.02 | 78.72 | 2.15 | 7.90 |
| BCP (Repr) | SGD | 4/76 | 88.27 (3.26) | 79.15 (5.01) | 2.26 (1.03) | 8.21 (4.18) |
| BCP | $SGD_{CO}$ | 4/76 | 88.37 (3.11) | 79.29 (4.81) | 2.19 (0.88) | 8.07 (3.97) |
| BCP (Ours) | $SGD_{HCO}$ | 4/76 | **88.83 (2.64)** | **79.99 (4.17)** | **2.19 (0.75)** | **6.55 (2.73)** |
| *Semi-Supervised with BCP (8 Labeled)* | | | | | | |
| BCP (Paper) | SGD | 8/72 | 89.62 | 81.31 | 1.76 | 6.81 |
| BCP (Repr) | SGD | 8/72 | 89.64 (3.55) | 81.41 (5.52) | 1.78 (0.86) | 6.93 (3.93) |
| BCP | $SGD_{CO}$ | 8/72 | 89.76 (2.93) | 81.54 (4.72) | 1.72 (0.70) | 6.84 (3.65) |
| BCP (Ours) | $SGD_{HCO}$ | 8/72 | **90.75 (2.17)** | **83.13 (3.63)** | **1.64 (0.58)** | **5.52 (2.26)** |

0.01, decayed by 10% every 2.5K iterations. Each batch consisted of 4 labeled and 4 unlabeled patches.

We kept the baseline pre-training weights and use the HCO optimizer ($SGD_{HCO}$) in the self-training phase. This isolates the effect of cautious gradient integration during pseudo-label-driven learning. Results were averaged for 2 random seeds. and statistical significance was assessed with a one-tailed paired Wilcoxon signed-rank test. Each run took approximately 8 hours, for a total compute budget of about 48 GPU-hours.

**Results.** With 8 labeled scans, replacing SGD with $SGD_{HCO}$ improved Dice from 89.64 to 90.75 ($p = 0.0016$) and Jaccard from 81.41 to 83.13 ($p = 0.0016$), while reducing HD95 by 20.3% ($6.93{\rightarrow}5.52$; $p = 0.0063$) and ASD by 7.9% ($1.78{\rightarrow}1.64$; $p = 0.2729$). At 4 labels, Dice rose from 88.27 to 88.83 ($p = 0.0379$) and Jaccard from 79.15 to 79.99 ($p = 0.0413$), with HD95 dropping from 8.21 to 6.55 ($p = 0.0086$). This simple optimizer change closes much of the gap to full supervision, recovering $\sim$45% of the Dice gap, $\sim$52% of the ASD gap, and $\sim$81% of the HD95 gap.

**Interpretation.** These gains suggest that the hierarchical trust mechanism in $SGD_{HCO}$ enables more reliable integration of unlabeled gradients, translating into sharper boundaries and fewer large segmentation errors. HCO recovers much of the performance lost to label scarcity with minimal architectural or computational overhead.

## 4.2 STUDY 2: BCP ON THE PANCREAS-CT DATASET

**Objective.** We evaluate $Adam_{HCO}$'s performance against both standard Adam and $Adam_{CO}$ within the BCP framework (Bai et al., 2023) on the Pancreas-CT dataset (Roth et al., 2015). This study tests whether HCO generalizes to a different optimizer (Adam) and a new anatomical target and imaging modality (abdominal CT).

**Dataset and protocol.** We follow the same pre-processing and two-phase training schedule as in Study 1. We used 12 labeled and 50 unlabeled volumes from the Pancreas-CT dataset (Roth et al., 2015). Each run took approximately six GPU hours. Statistical significance was assessed with a one-tailed paired Wilcoxon signed-rank test.

Table 2: Single-run pancreas segmentation performance using 12 labeled and 50 unlabeled scans. **VNet** and **BCP** results for Adam, $\text{Adam}_{\text{CO}}$, and $\text{Adam}_{\text{HCO}}$. Metrics: Dice/Jaccard (%), ASD (voxels), and HD95 (voxels), evaluated on 18 held-out cases. Baseline results from (Bai et al., 2023; Song & Wang, 2024; Zhao et al., 2024). The L/U column indicates the number of labeled / unlabeled scans used. Bold indicates the best result for each metric.

| Method | Optimizer | L/U | Dice ↑ | Jaccard ↑ | HD95 ↓ | ASD ↓ |
|---|---|---|---|---|---|---|
| VNet | Adam | 12/0 | 69.96 | 55.55 | 14.27 | 1.64 |
| VNet | Adam | 62/0 | 82.60 | 70.81 | 5.61 | 1.33 |
| BCP | Adam | 12/50 | 82.91 | 70.97 | 6.43 | 2.25 |
| BCP | $\text{Adam}_{\text{CO}}$ | 12/50 | 82.58 (5.37) | 70.65 (7.57) | 8.84 (9.24) | 3.11 (2.38) |
| BCP (Ours) | $\text{Adam}_{\text{HCO}}$ | 12/50 | **83.71 (4.94)** | **72.26 (7.00)** | **5.46 (3.99)** | **2.05 (1.38)** |
| AD-MT | SGD | 6/50 | 80.21 | 67.51 | 7.18 | 1.66 |
| AD-MT (Ours) | $\text{SGD}_{\text{HCO}}$ | 6/50 | **81.36** | **68.86** | **5.02** | **1.51** |
| AD-MT | SGD | 12/50 | 82.61 | 70.70 | 4.94 | 1.38 |
| AD-MT (Ours) | $\text{SGD}_{\text{HCO}}$ | 12/50 | **83.53** | **71.93** | **4.77** | **1.27** |

**Results.** $\text{Adam}_{\text{HCO}}$ consistently outperformed $\text{Adam}_{\text{CO}}$ across all metrics (Table 2). Dice improved from 82.58 to 83.71 ($p = 0.0152$), Jaccard from 70.65 to 72.26 ($p = 0.0134$), HD95 from 8.84 to 5.46 ($p = 0.0149$), and ASD from 3.11 to 2.05 ($p = 0.0152$). Statistical testing was conducted only between $\text{Adam}_{\text{CO}}$ and $\text{Adam}_{\text{HCO}}$; standard Adam is reported for reference from (Bai et al., 2023) without variance estimates.

**Interpretation.** By boosting performance on abdominal CT segmentation with Adam, HCO demonstrates that its gains are not confined to cardiac MRI or SGD. Such robustness across optimizer, modality, and organ highlights its broad applicability to semi-supervised learning.

To further assess cross-framework generalization, we integrate HCO into the AD-MT (Zhao et al., 2024) framework using its official implementation and unchanged hyperparameters (Table 2). In both the 6/50 and 12/50 label regimes, replacing SGD with $\text{SGD}_{\text{HCO}}$ improved all segmentation metrics. This demonstrates that HCO's gains extend to another SSL paradigm without requiring framework-specific tuning.

### 4.3 STUDY 3: CMT ON THE FETAL-MRI DATASET

**Objective.** Evaluate whether HCO generalizes across semi-supervised learning frameworks, optimizer types, and anatomical targets by applying $\text{AdamW}_{\text{HCO}}$ within the consistency-based CMT framework (Shen et al., 2023) for fetal lung and liver segmentation.

**Training protocol.** Following CMT (Shen et al., 2023), we employ a 3D V-Net backbone (Milletari et al., 2016). Data augmentation consists of random flips and 3D crops. Each training iteration randomly samples eight patches of size $144 \times 144 \times 64$ (4 labeled, 4 unlabeled). AdamW and its HCO variant are used with a fixed learning rate of $1 \times 10^{-4}$. Training runs for 200 epochs. To avoid bias from a single labeled/unlabeled split we generate $4 \times 4 = 16$ combinations: four labeled subsets ($\text{L}_0-\text{L}_3$, 5 scans each) and four unlabeled subsets ($\text{U}_0-\text{U}_3$, 150 scans each). CMT is trained from scratch for every pair $(\text{L}_i, \text{U}_j)$. Each run took approximately 4 hours, for a total compute budget of about 256 GPU-hours.

**Results.** For each test volume we first averaged segmentation metrics over the 16 folds, and then computed the overall mean and standard deviation across all cases. Statistical significance was determined with a one-sided paired $t$-test. Table 3 lists the resulting segmentation metrics. For lungs, $\text{AdamW}_{\text{HCO}}$ demonstrated substantial improvements over AdamW: Dice increased by 24.6% (67.54→84.17), Jaccard increased by 39.3% (52.80→73.54), ASD decreased by 56.2% (8.12→3.56), and HD95 decreased by 47.0% (25.09→13.30). For liver, $\text{AdamW}_{\text{HCO}}$ also showed significant gains over AdamW: Dice increased by 15.7% (70.99→82.15), Jaccard increased by 25.3% (56.10→70.32), ASD decreased by 54.9% (10.95→4.94), and HD95 decreased by 52.2%

Table 3: Fetal MRI segmentation results (mean (std)) for liver and lungs. **AdamW$_{sup}$**: supervised baseline with 5 labeled scans. **AdamW**: standard semi-supervised baseline. **AdamW$_{CO}$**: Cautious Optimizer variant (Liang et al., 2024). **AdamW$_{HCO}$**: full Hierarchical Cautious Optimizer (ours). Metrics: Dice/Jaccard (%), Average Surface Distance (ASD, voxels), and 95th percentile Hausdorff Distance (HD95, voxels). Bold highlights the best result for each organ and metric.

| Organ | Optimizer | Metrics | | | |
|---|---|---|---|---|---|
| | | Dice ↑ | Jaccard ↑ | ASD ↓ | HD95 ↓ |
| Liver | AdamW$_{sup}$ | 61.09 (3.66) | 47.82 (3.88) | 7.15 (3.39) | 21.79 (6.73) |
| | AdamW | 70.99 (8.57) | 56.10 (9.88) | 10.95 (4.15) | 31.80 (10.54) |
| | AdamW$_{CO}$ | 63.36 (9.60) | 47.48 (9.98) | 13.53 (4.15) | 37.02 (9.45) |
| | AdamW$_{HCO}$ | **82.15 (5.40)** | **70.32 (7.49)** | **4.94 (2.27)** | **15.21 (6.91)** |
| Lungs | AdamW$_{sup}$ | 51.06 (8.94) | 40.75 (8.73) | 9.11 (0.67) | 26.22 (2.95) |
| | AdamW | 67.54 (10.24) | 52.80 (11.93) | 8.12 (3.83) | 25.09 (10.25) |
| | AdamW$_{CO}$ | 56.96 (10.71) | 41.38 (10.80) | 12.29 (4.49) | 34.52 (10.00) |
| | AdamW$_{HCO}$ | **84.17 (6.70)** | **73.54 (9.73)** | **3.56 (2.68)** | **13.30 (10.01)** |

(31.80→15.21). All these listed improvements for lung and liver segmentation metrics over AdamW were statistically significant (paired $t$-test, $p < 1 \times 10^{-18}$).

Figure 2 illustrates that AdamW$_{HCO}$ produced sharper boundaries and reduced false negatives in low-contrast regions.

**Interpretation.** These results demonstrate that AdamW$_{HCO}$ consistently improved segmentation performance across multiple organs, highlighting its ability to generalize beyond task-specific tuning. The gains in both lung and liver structures, despite their differing anatomical and contrast characteristics, suggest that HCO's hierarchical refinement and cautious gating robustly enhanced learning from limited labels. Combined with consistent improvements over standard AdamW, this supports the broader applicability of HCO as a general-purpose optimizer for semi-supervised segmentation.

### 4.4 Study 4: Hierarchical Processing Ablation

**Objective.** Evaluate the contribution of HCO's hierarchical processing by comparing against the original Cautious Optimizer (CO) (Liang et al., 2024) across all three experimental settings.

**Results.** Tables 1, 2, and 3 consistently demonstrate that CO's gradient rejection approach underperforms both standard optimizers and HCO across diverse settings:

**Left-Atrium (SGD):** CO achieves 89.76% Dice vs. SGD's 89.64% and HCO's 90.75%—showing minimal gains over the standard optimizer, while HCO yields a clear improvement.

**Pancreas (Adam):** CO substantially underperforms with 82.57% Dice vs. Adam's 82.91% and HCO's 83.71%, along with worse surface metrics (ASD: 9.21 vs. 6.43 vs. 5.46).

**Fetal MRI (AdamW):** CO performs worst across both organs—lung Dice of 56.96% vs. AdamW's 67.54% vs. HCO's 84.17%; liver Dice of 63.36% vs. 70.99% vs. 82.15%.

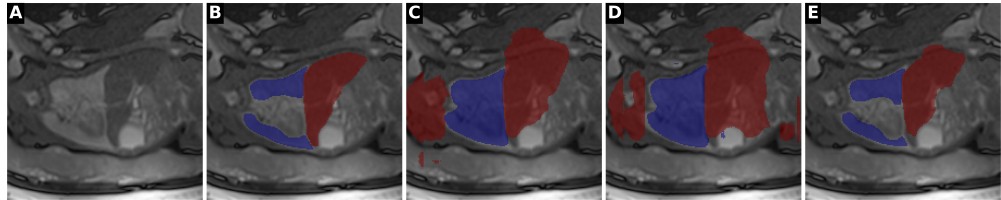

Figure 2: **A.** Input MRI, **B.** Ground truth, **C.** AdamW, **D.** AdamW$_{CO}$, **E.** AdamW$_{HCO}$(Ours). Lungs are shown in blue and liver in red.

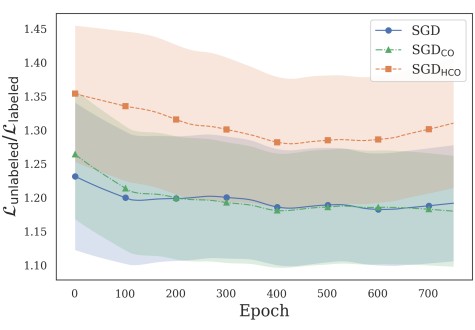

(a) Left Atrium / BCP (*SGD* family)

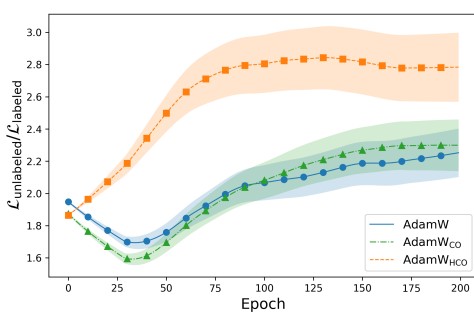

(b) Fetal MRI / CMT (*AdamW* family)

Figure 3: **Evolution of the unlabeled-to-labeled loss ratio.** Each curve is a LOWESS trend line of $\mathcal{L}_{\text{unlabeled}}/\mathcal{L}_{\text{labeled}}$ across epochs; shaded bands show $\pm$SEM computed across steps within each epoch. **(a)** On the LA dataset, $\text{SGD}_{\text{HCO}}$ (orange) maintains a higher $\mathcal{L}_{\text{unlabeled}}/\mathcal{L}_{\text{labeled}}$ throughout training than standard SGD (blue). **(b)** On fetal MRI, $\text{AdamW}_{\text{HCO}}$ (orange) exhibits a higher ratio than vanilla AdamW (blue) and the caution-only variant (orange).

**Interpretation.** Across all settings, our hierarchical method (HCO) **consistently and significantly outperforms standard optimizers**, demonstrating its effectiveness. In contrast, applying the same cautious principle without a hierarchy (CO) fails to provide meaningful benefits and may degrades performance. This highlights the critical role of hierarchy in HCO's effectiveness.

### 4.5 STUDY 5 — IMPACT OF HCO ON SUPERVISED–UNSUPERVISED LOSS DYNAMICS

**Objective.** Assess how HCO influences the balance between supervised and unsupervised loss contributions during training. We hypothesize that HCO variants sustain stronger reliance on unlabeled signals compared to their baseline counterparts.

**Protocol.** For each optimizer, we first compute the ratio ($\mathcal{L}_{\text{unlabeled}}/\mathcal{L}_{\text{labeled}}$) at every training step $t$. To summarize variability, we aggregate these ratios within each epoch to obtain the standard error of the mean (SEM) across the epoch. For BCP, independent runs correspond to different random seeds with a fixed labeled/unlabeled split, whereas for CMT they correspond to different labeled/unlabeled splits with a fixed seed. For visualization, we then fit a LOWESS trendline to the raw step-wise ratios. We evaluate three variants: AdamW (baseline), $\text{AdamW}_{\text{CO}}$ (Liang et al., 2024), and $\text{AdamW}_{\text{HCO}}$ on the fetal MRI task using the CMT framework; and two variants: SGD (baseline) and $\text{SGD}_{\text{HCO}}$ on the Left Atrium task using the BCP framework.

**Results.** Figure 3 shows that HCO consistently increases the relative contribution of the unlabeled loss over the course of training. On both tasks, the ratio $\mathcal{L}_{\text{unlabeled}}/\mathcal{L}_{\text{labeled}}$ remains higher under HCO than under the corresponding baseline. In the LA/BCP setting, $\text{SGD}_{\text{HCO}}$ yields an unlabeled-to-labeled ratio that is approximately 10% larger on average. A similar increase is observed for $\text{AdamW}_{\text{HCO}}$ in the Fetal-MRI/CMT setting. Curves for unlabeled loss and validation Dice are provided in Appendix D.

**Interpretation.** The observed dynamics indicate that HCO's gating and hierarchical momentum shift training toward greater utilization of unlabeled supervision, reflected by persistently higher $\mathcal{L}_{\text{unlabeled}}/\mathcal{L}_{\text{labeled}}$ ratios.

## 5 CONCLUSION

We presented HCO, a novel optimization framework that assigns trust only to labeled gradients while cautiously integrating aligned unlabeled signals. This approach is agnostic to network architecture and data modality, and we provide a formal proof that it preserves the convergence guarantees of its base optimizers. Together with consistent empirical gains and minimal code changes, these properties make HCO a strong default optimizer for low-label SSL regimes.

**Limitations.** Although HCO demonstrates significant potential, our empirical evaluation is presently restricted to specific segmentation datasets and semi-supervised learning frameworks. Additionally, the effectiveness of HCO inherently depends on the quality of the limited labeled data used to establish the trusted momentum direction. The gradient alignment mechanism and its theoretical justification rely on specific assumptions, which should be tested further through both empirical studies and theoretical analysis.

**Reproducibility statement** We provide complete pseudocode for our proposed optimizers (Algorithms 1–2) and a formal convergence proof in Appendix A. Dataset acquisition, annotation, and inter-observer details are given in Appendix B, while training protocols, hyperparameters, and optimization settings are described in Section 4 or follow the original BCP (Bai et al., 2023) and CMT (Shen et al., 2023) implementations for fair comparison. We ran multiple independent seeds and labeled/unlabeled splits to assess variability and report statistical tests. An anonymized code release will be included in the supplementary material to facilitate exact reproduction.

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

## A  CONVERGENCE PROOF DETAILS FOR HCO

Hierarchical Cautious Optimization (HCO) is designed for semi-supervised learning (SSL). Unlike standard optimizers such as Adam (Kingma & Ba, 2015) which treat labeled and unlabeled gradients equally when applied to a combined loss, HCO establishes a trust hierarchy. It computes momentum and variance estimates primarily from labeled data ($g_t^L$) and selectively incorporates gradients from unlabeled data ($g_t^U$) only when they align with the momentum direction derived from labeled data. This section provides a theoretical analysis of HCO's convergence properties. We demonstrate that under standard assumptions (Reddi et al., 2019) and a key assumption regarding HCO's cautious mechanism, HCO converges to a stationary point of a combined objective function.

### A.1  PRELIMINARIES AND ASSUMPTIONS

We consider the minimization of the composite SSL loss function:

$$\mathcal{L}(w) = \alpha \mathcal{L}_{\mathrm{L}}(w) + (1 - \alpha)\mathcal{L}_{\mathrm{U}}(w)$$

where $w \in \mathbb{R}^d$ are the model parameters, $\mathcal{L}_{\mathrm{L}}$ and $\mathcal{L}_{\mathrm{U}}$ are the losses on labeled and unlabeled data respectively, and $\alpha \in [0, 1]$ is a balancing coefficient. $\nabla \mathcal{L}_{\text{labeled}}(w)$, $\nabla \mathcal{L}_{\text{unlabeled}}(w)$, and $\nabla \mathcal{L}(w)$ denote the true gradients, while $g_t^L$ and $g_t^U$ are stochastic gradient estimates obtained at iteration $t$ from mini-batches for parameters $w_{t-1}$. Let $\mathcal{F}_{t-1}$ be the sigma-algebra generated by the random variables up to iteration $t - 1$ (i.e., $w_0, \ldots, w_{t-1}$ and any randomness used to compute them). Our analysis relies on the following assumptions:

**Assumption 1** (L-Smoothness). *The individual loss functions $\mathcal{L}_L(w)$ and $\mathcal{L}_U(w)$ are differentiable and $L_i$-smooth for $i \in \{L, U\}$, respectively. That is, for some constants $L_L, L_U > 0$, their gradients are Lipschitz continuous:*

$$||\nabla \mathcal{L}_i(w_1) - \nabla \mathcal{L}_i(w_2)|| \leq L_i ||w_1 - w_2||, \quad \forall w_1, w_2 \in \mathbb{R}^d$$

*This implies the combined loss $\mathcal{L}(w)$ is also L-smooth with $L = \alpha L_L + (1 - \alpha)L_U$.*

**Assumption 2** (Unbiased Stochastic Gradients). *The stochastic gradients $g_t^L$ and $g_t^U$ are unbiased estimators of the true gradients at $w_{t-1}$, conditioned on the history $\mathcal{F}_{t-1}$:*

$$\mathbb{E}[\,g_t^L \,|\, \mathcal{F}_{t-1}] = \nabla\mathcal{L}_{labeled}(w_{t-1}), \quad \mathbb{E}[\,g_t^U \,|\, \mathcal{F}_{t-1}] = \nabla\mathcal{L}_{unlabeled}(w_{t-1})$$

*This implies the combined stochastic gradient (if it were formed as $g_t = \alpha g_t^L + (1-\alpha)g_t^U$) would be an unbiased estimator of $\nabla\mathcal{L}(w_{t-1})$. HCO uses $g_t^L$ and $g_t^U$ in a structured, conditional way.*

**Assumption 3** (Bounded Variance). *The variance of the stochastic gradients is uniformly bounded by some constants $\sigma_L^2 \geq 0$ and $\sigma_U^2 \geq 0$:*

$$\mathbb{E}[\,||g_t^L - \nabla\mathcal{L}_{labeled}(w_{t-1})||^2 \,|\, \mathcal{F}_{t-1}] \leq \sigma_L^2$$

$$\mathbb{E}[\,||g_t^U - \nabla\mathcal{L}_{unlabeled}(w_{t-1})||^2 \,|\, \mathcal{F}_{t-1}] \leq \sigma_U^2$$

*This implies that $\mathbb{E}[\,||g_t^L||^2 \,|\, \mathcal{F}_{t-1}] \leq ||\nabla\mathcal{L}_{labeled}(w_{t-1})||^2 + \sigma_L^2$ and similarly for $g_t^U$. If the true gradients are bounded (e.g., if $\mathcal{L}$ is further assumed to be non-convex but a global Lipschitz constant exists for the gradient, or we analyze convergence in a bounded domain), then the stochastic gradients are also bounded in the second moment.*

**Assumption 4** (Step Size Conditions). *The learning rates $\varepsilon_t > 0$ satisfy:*

$$\sum_{t=1}^{\infty} \varepsilon_t = \infty \quad and \quad \sum_{t=1}^{\infty} \varepsilon_t^2 < \infty$$

*This is typically satisfied by the schedules $\varepsilon_t = \eta/\sqrt{t}$ for a base rate $\eta > 0$.*

**Assumption 5** (Bounded Moments and Update Directions). *Under Assumptions 2 and 3, the expected norms of the first moment estimates $m_t$ and element-wise second moment estimates $v_t$ (as computed in Algorithm 1) are bounded. Consequently, the update directions $u_t^L$ (derived from labeled data) and $u_t$ (derived from conditionally combined data, see algorithm line 18) are bounded in expectation. The elements of $v_t$ are non-negative due to squaring, ensuring denominators in $u_t^L$ and $u_t$ (e.g., $\sqrt{(\hat{v}_t)_i} + \epsilon$) are bounded below by $\epsilon > 0$. The scaled mask $\overline{\phi}_t$ is also bounded element-wise, $0 \leq (\overline{\phi}_t)_i \leq 1/\epsilon'$ (where $\epsilon'$ is the $\epsilon$ in $\max(mean(\phi_t), \epsilon)$ from algorithm line 21). This implies that the total update direction $\Delta_t = u_t^L + \overline{\phi}_t \odot u_t$ has bounded expected squared norm, i.e., $\mathbb{E}[\,||\Delta_t||^2 \,|\, \mathcal{F}_{t-1}] \leq B^2$ for some constant $B > 0$. This relies on $m_t, v_t$ being convex combinations of initial moments and (squared) gradients, which are themselves assumed to have bounded variance. The bias correction terms $(1 - \beta_1^t)$ and $(1 - \beta_2^t)$ are well-behaved.*

**Remark 1** (On Conditional Update Term and HCO Mechanism). *The convergence analysis critically relies on the behavior of the update term $\Delta_t$. The component $u_t^L$ is derived from labeled data and provides a trusted update direction. The term $\overline{\phi}_t \odot u_t$ incorporates unlabeled data $g_t^U$ conditionally. The mask $\phi_t = \mathbb{I}(u_t^L \odot g_t^U > 0)$ aims to ensure that $g_t^U$ components are included only when they align with established momentum directions ($u_t^L$ for moment updates, $u_t$ for the final update step).*

*The core assumption for HCO's convergence, as stated in Theorem 1, is that this cautious mechanism is effective. Specifically, we assume that in expectation, the conditional inclusion of unlabeled gradient information via $\overline{\phi}_t \odot u_t$ provides a non-negative or sufficiently small negative contribution to the overall descent with respect to the combined loss $\mathcal{L}(w)$. That is, $\mathbb{E}[\,\langle\nabla\mathcal{L}(w_{t-1}), \overline{\phi}_t \odot u_t\rangle \,|\, \mathcal{F}_{t-1}]$ does not significantly counteract the descent from the labeled term. This assumption is plausible in SSL contexts where pseudo-labels (often driving $\mathcal{L}_U$) are intended to approximate true labels, and HCO's alignment filtering is designed to select for such agreement. **We provide empirical validation of this alignment behavior in Appendix E.***

*The structure of $v_t$ in HCO is also notable: it is first derived from $(g_t^L)^2$ and then augmented by $(g_t^U \odot \phi_t)^2$. This mixed $v_t$ is used to normalize $m_t$ (which is similarly mixed) to form $u_t$. This adaptive normalization, using variance information from both sources (conditionally), is a feature of HCO. Assumption 5 implies that this structure still leads to well-behaved update directions.*

PROOF STRATEGY FOR HCO CONVERGENCE

The convergence proof for HCO is developed within the established framework for stochastic gradient methods, particularly drawing analogies from analyses of adaptive optimizers like

Adam/AdamW (Kingma & Ba, 2015; Loshchilov & Hutter, 2019; Reddi et al., 2019). The general approach involves utilizing a Lyapunov argument based on the objective function $\mathcal{L}(w)$ to demonstrate expected descent or bounded progress.

Our analysis leverages several standard elements common to such proofs:

- We employ typical assumptions in stochastic optimization: L-smoothness of the objective function (Assumption 1), unbiased stochastic gradient estimates (Assumption 2), bounded variance of these gradients (Assumption 3), and conventional decaying step-size conditions (Assumption 4).

- Similar to other adaptive methods, we assume the boundedness of the optimizer's internal moment estimates and the resultant update directions (Assumption 5), crucial for ensuring the stability of the updates.

- The macroscopic structure of the proof follows a familiar path: bounding the expected one-step change in the loss function, summing these bounds telescopically over iterations, and then utilizing the step-size properties to establish the convergence of gradient norms.

The main distinction and analytical challenge in proving HCO's convergence stem from its unique, hierarchically structured update rule for $\Delta_t = u_t^L + \overline{\phi}_t \odot u_t$. Unlike optimizers that aggregate gradients more directly, HCO's mechanism for incorporating gradients from unlabeled data ($g_t^U$) is conditional, depending on alignment with trusted momentum derived primarily from labeled data ($g_t^L$).

Consequently, the central theoretical addition specific to HCO in this analysis is the core assumption within Theorem 1 (further elaborated in Remark 1). This assumption posits that despite HCO's conditional and composite update structure, the overall update direction $\Delta_t$ maintains a sufficiently strong positive correlation in expectation with the negative true gradient $-\nabla\mathcal{L}(w_{t-1})$ of the **combined** SSL objective. This is the pivotal hypothesis that allows HCO's specialized mechanism to integrate within the broader convergence theory: it asserts that HCO's cautious filtering is effective in guiding the optimization process without fundamentally undermining the descent properties required for convergence.

Once this HCO-specific descent condition (i.e., the lower bound on $\mathbb{E}[\langle \nabla\mathcal{L}(w_{t-1}), \Delta_t \rangle \,|\, \mathcal{F}_{t-1}]$) is established by this key assumption, the rest of the proof proceeds similarly to standard convergence analyses for stochastic adaptive optimizers (e.g., (Bottou et al., 2018; Reddi et al., 2019)), using telescoping sums and step-size conditions to derive rates. Our analysis thus highlights the critical condition under which HCO's distinct design is proven to converge.

## A.2 Convergence Analysis

We analyze the convergence of HCO (Algorithm 1) to a stationary point of the objective function $\mathcal{L}(w)$. For this analysis, we consider the parameter update rule $w_t = w_{t-1} - \varepsilon_t \Delta_t$, where $\Delta_t = u_t^L + \overline{\phi}_t \odot u_t$. The weight decay step is typically analyzed separately or considered part of $\mathcal{L}(w)$; its impact on the gradient norm convergence proof for Adam-like optimizers is often treated as a secondary effect for simplicity.

**Theorem 1** (Convergence of HCO). *Let Assumptions 1-5 hold. Assume further, as discussed in Remark 1, that the expected contribution from the conditional unlabeled term is sufficiently well-behaved such that the overall update direction $\Delta_t$ has a significant positive correlation in expectation with the negative gradient $-\nabla\mathcal{L}(w_{t-1})$. Specifically, assume there exist constants $C_1 > 0$ and an effective upper bound $V_{bound} > \epsilon$ for the components of $\mathbb{E}[\sqrt{\hat{v}_t} \,|\, \mathcal{F}_{t-1}]$ (where $\hat{v}_t$ refers to the relevant second moment estimates used in $\Delta_t$, implied by Assumption 5) such that the expected inner product satisfies:*

$$\mathbb{E}[\langle \nabla\mathcal{L}(w_{t-1}), \Delta_t \rangle \,|\, \mathcal{F}_{t-1}] \geq \frac{C_1}{V_{bound}} ||\nabla\mathcal{L}(w_{t-1})||^2$$

*Then, the sequence of iterates $\{w_t\}$ generated by the HCO algorithm with learning rates $\varepsilon_t$ satisfying Assumption 4 converges to a stationary point of $\mathcal{L}(w)$ in the sense that:*

$$\lim_{T \to \infty} \frac{1}{T} \sum_{t=1}^{T} \mathbb{E}[||\nabla\mathcal{L}(w_{t-1})||^2] = 0$$

*The convergence rate for $\min_{0 \leq t \leq T-1} \mathbb{E}[||\nabla\mathcal{L}(w_t)||^2]$ is expected to be $\mathcal{O}(1/\sum_{k=1}^{T} \varepsilon_k)$, which is $\mathcal{O}(1/\sqrt{T})$ if $\varepsilon_t \propto 1/\sqrt{t}$, similar to Adam.*

*Proof.* The proof follows the standard Lyapunov function approach for stochastic gradient methods. We consider the one-step progress in the objective function $\mathcal{L}(w)$. By Assumption 1 (L-smoothness of $\mathcal{L}$), we have:

$$\mathcal{L}(w_t) \leq \mathcal{L}(w_{t-1}) + \langle \nabla\mathcal{L}(w_{t-1}), w_t - w_{t-1} \rangle + \frac{L}{2}||w_t - w_{t-1}||^2$$

Substituting the HCO update $w_t - w_{t-1} = -\varepsilon_t \Delta_t$:

$$\mathcal{L}(w_t) \leq \mathcal{L}(w_{t-1}) - \varepsilon_t \langle \nabla\mathcal{L}(w_{t-1}), \Delta_t \rangle + \frac{L\varepsilon_t^2}{2}||\Delta_t||^2$$

Taking conditional expectation $\mathbb{E}[\,\cdot\,|\,\mathcal{F}_{t-1}]$ (expectation with respect to $\mathcal{F}_{t-1}$):

$$\mathbb{E}[\,\mathcal{L}(w_t)\,|\,\mathcal{F}_{t-1}] \leq \mathcal{L}(w_{t-1}) - \varepsilon_t \mathbb{E}[\,\langle \nabla\mathcal{L}(w_{t-1}), \Delta_t \rangle\,|\,\mathcal{F}_{t-1}] + \frac{L\varepsilon_t^2}{2}\mathbb{E}[\,||\Delta_t||^2\,|\,\mathcal{F}_{t-1}]$$

Under Assumption 5, the last term is bounded: $\mathbb{E}[\,||\Delta_t||^2\,|\,\mathcal{F}_{t-1}] \leq B^2$ for some constant $B > 0$. The core of the theorem lies in the assumption regarding the expected inner product term:

$$\mathbb{E}[\,\langle \nabla\mathcal{L}(w_{t-1}), \Delta_t \rangle\,|\,\mathcal{F}_{t-1}] \geq \frac{C_1}{V_{\text{bound}}}||\nabla\mathcal{L}(w_{t-1})||^2$$

Let $\tilde{C}_1 = C_1/V_{\text{bound}}$. Then $\tilde{C}_1 > 0$. Substituting these into the inequality for $\mathbb{E}[\,\mathcal{L}(w_t)\,|\,\mathcal{F}_{t-1}]$:

$$\mathbb{E}[\,\mathcal{L}(w_t)\,|\,\mathcal{F}_{t-1}] \leq \mathcal{L}(w_{t-1}) - \varepsilon_t \tilde{C}_1 ||\nabla\mathcal{L}(w_{t-1})||^2 + \frac{LB^2\varepsilon_t^2}{2}$$

Rearranging the terms to isolate the squared gradient norm:

$$\varepsilon_t \tilde{C}_1 ||\nabla\mathcal{L}(w_{t-1})||^2 \leq \mathcal{L}(w_{t-1}) - \mathbb{E}[\,\mathcal{L}(w_t)\,|\,\mathcal{F}_{t-1}] + \frac{LB^2\varepsilon_t^2}{2}$$

Taking the total expectation $\mathbb{E}[\cdot]$ over all randomness:

$$\varepsilon_t \tilde{C}_1 \mathbb{E}[||\nabla\mathcal{L}(w_{t-1})||^2] \leq \mathbb{E}[\mathcal{L}(w_{t-1})] - \mathbb{E}[\mathcal{L}(w_t)] + \frac{LB^2\varepsilon_t^2}{2}$$

Summing this inequality from $t = 1$ to $T$:

$$\tilde{C}_1 \sum_{t=1}^{T} \varepsilon_t \mathbb{E}[||\nabla\mathcal{L}(w_{t-1})||^2] \leq \sum_{t=1}^{T} (\mathbb{E}[\mathcal{L}(w_{t-1})] - \mathbb{E}[\mathcal{L}(w_t)]) + \frac{LB^2}{2} \sum_{t=1}^{T} \varepsilon_t^2$$

The first sum on the right-hand side is a telescoping sum:

$$\sum_{t=1}^{T} (\mathbb{E}[\mathcal{L}(w_{t-1})] - \mathbb{E}[\mathcal{L}(w_t)]) = \mathbb{E}[\mathcal{L}(w_0)] - \mathbb{E}[\mathcal{L}(w_T)]$$

Assuming the loss function is bounded below by $\mathcal{L}^*$, i.e., $\mathcal{L}(w) \geq \mathcal{L}^*$ for all $w$:

$$\mathbb{E}[\mathcal{L}(w_0)] - \mathbb{E}[\mathcal{L}(w_T)] \leq \mathcal{L}(w_0) - \mathcal{L}^*$$

So, we have:

$$\tilde{C}_1 \sum_{t=1}^{T} \varepsilon_t \mathbb{E}[||\nabla\mathcal{L}(w_{t-1})||^2] \leq \mathcal{L}(w_0) - \mathcal{L}^* + \frac{LB^2}{2} \sum_{t=1}^{T} \varepsilon_t^2$$

By Assumption 4, $\sum_{t=1}^{\infty} \varepsilon_t^2 < \infty$. Let $S_2 = \sum_{t=1}^{\infty} \varepsilon_t^2$. The right-hand side is bounded by a constant:

$$\sum_{t=1}^{T} \varepsilon_t \mathbb{E}[||\nabla\mathcal{L}(w_{t-1})||^2] \leq \frac{1}{\tilde{C}_1} \left( \mathcal{L}(w_0) - \mathcal{L}^* + \frac{LB^2 S_2}{2} \right) := K_{\text{const}}$$

Let $G_t = \mathbb{E}[||\nabla\mathcal{L}(w_t)||^2]$. We have $\sum_{t=1}^{T} \varepsilon_t G_{t-1} \leq K_{\text{const}}$. Since $G_{t-1} \geq 0$ and $\varepsilon_t > 0$:

$$\min_{0 \leq \tau \leq T-1} G_\tau \sum_{t=1}^{T} \varepsilon_t \leq \sum_{t=1}^{T} \varepsilon_t G_{t-1} \leq K_{\text{const}}$$

Therefore,

$$\min_{0 \leq \tau \leq T-1} \mathbb{E}[||\nabla\mathcal{L}(w_\tau)||^2] \leq \frac{K_{\text{const}}}{\sum_{t=1}^{T} \varepsilon_t}$$

By Assumption 4, $\sum_{t=1}^{\infty} \varepsilon_t = \infty$. Thus, as $T \to \infty$, the right-hand side goes to 0. This implies that $\liminf_{T \to \infty} \mathbb{E}[||\nabla\mathcal{L}(w_T)||^2] = 0$.

We now turn to the convergence of the average squared gradient norm, a standard criterion in optimization analysis:

$$\frac{1}{T} \sum_{t=1}^{T} \mathbb{E}[||\nabla\mathcal{L}(w_{t-1})||^2]$$

Let $G_t = \mathbb{E}[||\nabla\mathcal{L}(w_t)||^2]$. We have established the inequality:

$$\sum_{t=1}^{T} \varepsilon_t G_{t-1} \leq K_{\text{const}}$$

Assuming the learning rates $\varepsilon_t$ are positive and non-increasing (a common property, e.g., for $\varepsilon_t = \eta/\sqrt{t}$), it holds that $\varepsilon_T \leq \varepsilon_t$ for $1 \leq t \leq T$. Thus, $\varepsilon_T$ is the minimum learning rate in this range up to $T$. We can then bound the sum of gradients:

$$\varepsilon_T \sum_{t=1}^{T} G_{t-1} \leq \sum_{t=1}^{T} \varepsilon_t G_{t-1} \leq K_{\text{const}}$$

Dividing by $T\varepsilon_T$ (since $\varepsilon_T > 0$ by Assumption 4):

$$\frac{1}{T} \sum_{t=1}^{T} G_{t-1} \leq \frac{K_{\text{const}}}{T\varepsilon_T}$$

Now, consider the specific learning rate schedule $\varepsilon_t = \eta/\sqrt{t}$ for some constant $\eta > 0$. For this schedule, $\varepsilon_T = \eta/\sqrt{T}$. The term $T\varepsilon_T$ in the denominator becomes:

$$T\varepsilon_T = T\left(\frac{\eta}{\sqrt{T}}\right) = \eta\sqrt{T}$$

Substituting this into our inequality for the average squared gradient norm:

$$\frac{1}{T} \sum_{t=1}^{T} \mathbb{E}[||\nabla\mathcal{L}(w_{t-1})||^2] \leq \frac{K_{\text{const}}}{\eta\sqrt{T}}$$

As $T \to \infty$, the right-hand side $\frac{K_{\text{const}}}{\eta\sqrt{T}}$ clearly tends to 0. This completes the proof that

$$\lim_{T \to \infty} \frac{1}{T} \sum_{t=1}^{T} \mathbb{E}[||\nabla\mathcal{L}(w_{t-1})||^2] = 0$$

Finally, we summarize the convergence rate for the minimum expected squared gradient norm, which was derived as:

$$\min_{0 \leq \tau \leq T-1} \mathbb{E}[||\nabla\mathcal{L}(w_\tau)||^2] \leq \frac{K_{\text{const}}}{\sum_{t=1}^{T} \varepsilon_t}$$

This provides a general convergence rate of $\mathcal{O}\left(1/\sum_{k=1}^{T} \varepsilon_k\right)$. For the specific case where $\varepsilon_t = \eta/\sqrt{t}$, the sum of learning rates can be approximated for large $T$ as $\sum_{k=1}^{T} \varepsilon_k \approx 2\eta\sqrt{T}$. Substituting this approximation, the convergence rate for the minimum expected squared gradient norm becomes $\mathcal{O}(1/\sqrt{T})$.

$\square$

### A.3 DISCUSSION

The convergence analysis establishes that HCO, under standard assumptions for stochastic gradient methods plus a key assumption on the effectiveness of its cautious mechanism (Theorem 1 and Remark 1), converges to a stationary point of the combined SSL objective $\mathcal{L}(w)$. The convergence rate is comparable to standard Adam when similar step size schedules are used. The theoretical justification relies on the labeled gradient $g_t^L$ driving the primary update direction, while the unlabeled gradient $g_t^U$ is incorporated conditionally based on alignment. This design aims to achieve robustness against noisy pseudo-labels in SSL without sacrificing fundamental convergence guarantees. The crucial assumption is that this conditional alignment mechanism effectively ensures that the total update direction correlates positively with the negative true gradient of the overall loss function.

## B  FETAL MRI DATASET CONSTRUCTION

We curated a fetal MRI segmentation dataset to evaluate the robustness of semi-supervised optimization in challenging clinical conditions. The dataset comprises retrospective fetal body MRI scans collected at [XX Medical Center ] ([City, Country/State] annonimized), covering gestational ages between 28 and 39 weeks. All scans were acquired using True Fast Imaging with Steady-State Free Precession (TRUFI) sequences on 3T Siemens scanners (Prisma, Vida, or Skyra), utilizing an 18-channel body coil.

**Annotation protocol.** Ground-truth segmentations for the liver and lungs were obtained via a multi-stage annotation pipeline designed to balance annotation quality with radiologist time. Initially, 10 cases for lungs and 15 for liver were manually annotated from scratch by two expert radiologists to assess inter-observer variability and establish a reference set. These annotations were used to train a coarse-resolution CMT (Shen et al., 2023) model (resampled to $3 \times 3 \times 3$ mm), which generated preliminary segmentations to guide the extraction of 692 cropped sub-volumes ($180 \times 180 \times 80$ voxels) from full-resolution scans. A second-stage CMT model was then trained on the sub-volumes at the original resolution ($0.78 \times 0.78 \times 2$ mm), using a sliding window approach (patch size $144 \times 144 \times 64$) to manage GPU memory constraints. To produce the final training labels, 92 outputs from this model were randomly sampled and manually corrected by a clinical expert. This two-tiered approach yielded a high-quality curated dataset, balancing annotation fidelity with efficient use of radiologist time.

**Inter-observer variability.** Agreement between expert annotators was assessed using the Dice score, the Average Symmetric Surface Distance (ASD), and the 95th percentile Hausdorff distance (HD95). For the liver ($n = 15$), agreement yielded a Dice of $0.86 \pm 0.03$, ASD of $1.52 \pm 0.38$ mm, and HD95 of $4.82 \pm 1.36$ mm. For the lungs ($n = 10$), the Dice was $0.86 \pm 0.02$, with ASD $0.97 \pm 0.34$ mm and HD95 of $3.69 \pm 2.40$ mm. These metrics establish a realistic baseline for inter-observer agreement in fetal MRI segmentation and serve as a benchmark for future improvements.

### B.1 LICENSES

This section details the licenses associated with the datasets and key software components referenced in this work. Users should always refer to the original source of any dataset or software for the most accurate and complete licensing information.

- **BCP** The BCP framework utilized in this research is available under the MIT License.

- **Left Atrium Dataset (2018 Atria Segmentation Challenge):** The 2018 Atria Segmentation Challenge dataset (Xiong et al., 2021) was used in this research. The dataset comprises 154 3D late gadolinium-enhanced MRI (LGE-MRI) scans, with 100 cases provided with segmentation labels for training purposes. While the dataset is publicly available for academic research via the challenge organizers, explicit licensing terms were not specified with the data distribution. Our use of the dataset adheres to its intended purpose for academic research and comparison.

- **Pancreas-CT NIH Dataset (TCIA):** The NIH Pancreas-CT dataset was obtained from The Cancer Imaging Archive (TCIA) (Roth et al., 2015). This dataset is released under the Creative Commons Attribution 3.0 Unported (CC BY 3.0) license. This license permits use, distribution, and

adaptation of the dataset for both academic and commercial purposes, provided that appropriate credit is given to the original authors and TCIA. Appropriate attribution has been provided as per the license terms.

- **CMT (Collaborative Mean Teacher) Model Architecture:** The CMT, as described in (Shen et al., 2023), was used as a basis for model development in our annotation pipeline and optimizer comparison. The original authors of CMT did not explicitly disclose a software license for their reference implementation at the time of our inquiry, and attempts to clarify via direct contact within a reasonable timeframe were unsuccessful. Our use of the architectural principles is for research purposes.

## C  HCO TIME OVERHEAD

All wall-clock times in Table 4 were measured on the fetal MRI segmentation task using the collaborative mean teacher (CMT) framework (200 epochs, identical data splits and hyperparameters, single GPU). Our full Hierarchical Cautious Optimizer (AdamW$_{HCO}$) incurs less than a 1 % training-time overhead compared to standard AdamW.

Table 4: Training time comparison (wall-clock hours). Mean (std) over 16 runs.

| Optimizer | Training time (h) |
|---|---|
| AdamW | 4.16 (0.18) |
| AdamW$_{CO}$ | 4.17 (0.21) |
| AdamW$_{HCO}$ | 4.18 (0.22) |

# D    EXTENDED ANALYSIS OF LOSS BEHAVIOR ACROSS TRAINING

(a) Left Atrium / BCP — Labeled loss

(b) Left Atrium / BCP — Unlabeled loss

(c) Fetal MRI / CMT — Labeled loss

(d) Fetal MRI / CMT — Unlabeled loss

Figure 4: **Training dynamics of labeled and unlabeled losses.** Each curve is a LOWESS trend line of the labeled or unlabeled loss across epochs; shaded bands denote $\pm$SEM computed across steps within each epoch. **Top:** BCP framework on the Left Atrium dataset. **Bottom:** CMT framework on the Fetal MRI dataset.

## E    EMPIRICAL VALIDATION OF ALIGNMENT ASSUMPTIONS

To validate the convergence assumptions in Remark 1, we track the *cosine similarity* between labeled and unlabeled gradients, $\text{sim}(g_t^L, g_t^U) = \langle g_t^L, g_t^U \rangle / (\|g_t^L\| \|g_t^U\|)$, throughout training of the BCP baseline (Table 1). Under the distributional consistency hypothesis, $g_t^L$ serves as a proxy for the true gradient $\nabla \mathcal{L}_{\text{true}}$. Figure 5 shows the resulting distribution is approximately Gaussian with a slight positive skew (mean $\approx 0.03$).

**Relevance to Convergence Analysis**   The positive mean confirms that labeled and unlabeled signals generally align in expectation. However, the significant negative tail reveals frequent steps where noisy pseudo-labels directly contradict the true descent direction. HCO acts as a *gradient rectifier* by explicitly truncating this negative tail. This guarantees that the expected alignment of the final update, $\mathbb{E}[\langle \nabla \mathcal{L}_{\text{true}}, \Delta_t \rangle]$, is strictly superior to that of the raw gradients, preventing the auxiliary term from counteracting the trusted descent.

**Distinction from Cautious Optimizers.**   This analysis demonstrates that optimization conflicts originate specifically between data sources ($L$ vs $U$). Standard cautious methods, which only check the *total* gradient, miss this distinction. HCO's *hierarchical* approach is therefore necessary to specifically target and filter these source-level conflicts using the trusted $g^L$-derived momentum.

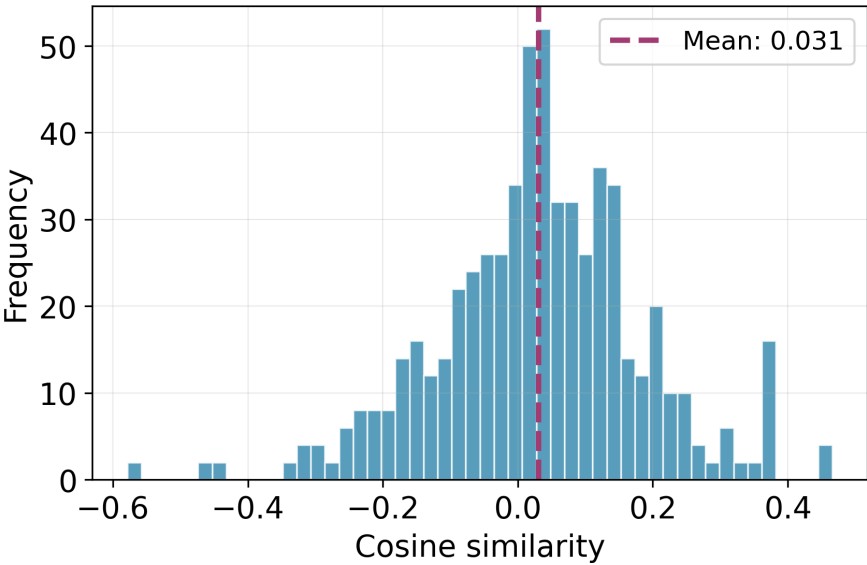

Figure 5: **Distribution of cosine similarity between labeled and unlabeled gradients.** Histogram of the per-step cosine similarity $\text{sim}(g_t^L, g_t^U)$ computed during training using the BCP framework on the Left Atrium dataset. The vertical dashed line indicates the empirical mean (0.03). While the positive mean suggests general alignment, the substantial probability mass in the negative region reveals frequent conflicts where noisy pseudo-labels contradict the trusted labeled signal. HCO's hierarchical masking effectively filters this negative tail, ensuring that unlabeled gradients are used only when they positively correlate with the trusted supervision.

## F  ADDITIONAL QUALITATIVE RESULTS

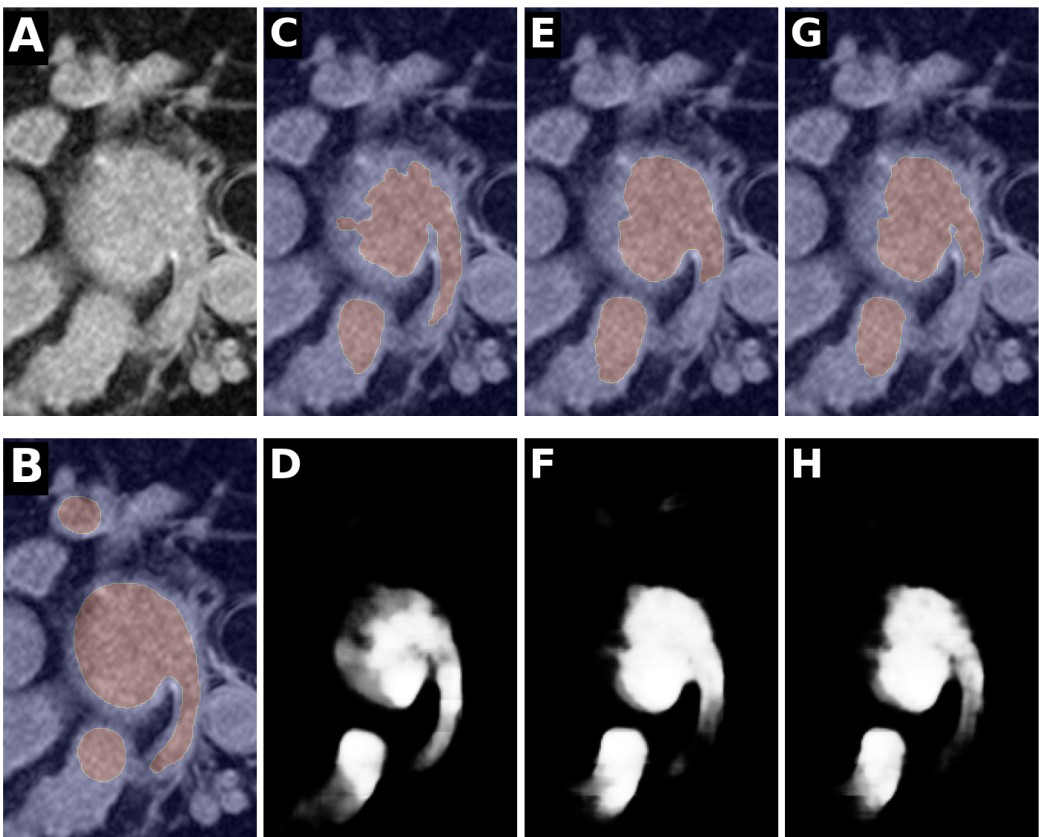

Figure 6: **Qualitative comparison on a difficult Left Atrial MRI case.** Examples are taken from the `BCP` framework trained with 8 labeled scans. (A) Raw MRI slice and (B) ground truth segmentation. Predictions (top) and corresponding probability maps (bottom) are shown for (C)–(D) $Adam_{HCO}$, (E)–(F) $Adam_{CO}$, and (G)–(H) standard Adam.

