# OpenReview forum: "Hierarchical Cautious Optimization for Semi-Supervised Medical Image Segmentation with Limited Labeled Data"
_ICLR.cc/2026/Conference — Submitted to ICLR 2026_

### Official Review · Reviewer_7dPQ · 2025-10-19

**Soundness:** 3
**Presentation:** 4
**Contribution:** 3
**Rating:** 4
**Confidence:** 5

**Summary:**

This paper addresses the challenge of noisy optimization in semi-supervised medical image segmentation, where limited annotations can lead to unreliable pseudo-labels. To mitigate this, the authors propose Hierarchical Cautious Optimization (HCO), an optimizer that establishes a trust hierarchy between labeled and unlabeled gradients. It is a kind of extension of previous Cautious Optimization work. The method is lightweight, generalizable to various optimizers (SGD, Adam), and introduces negligible computational overhead. Experiments across three datasets (Left Atrium MRI, Pancreas CT, and Fetal MRI) and two backbone optimizers demonstrate consistent improvements in Dice and surface metrics, confirming HCO’s effectiveness as a general-purpose optimization strategy for semi-supervised segmentation.

**Strengths:**

Addressing the noisy optimization problem is new and interesting in semi-supervised medical image segmentation.

The verified scenario is well-suited to the method designs.

The performance is consistently improved across different settings.

The paper is clear and well-written.

**Weaknesses:**

While the key idea of hierarchical cautious optimization is interesting, the technical novelty beyond prior work (Cautious Optimizers) could be further clarified. The paper may benefit from emphasizing medical-specific considerations, such as how anatomical or imaging characteristics influence gradient noise.

The experimental validation, although multi-dataset, is limited to two backbone optimizers. Given the wide variety of open-source SSL frameworks, additional and extensive comparisons could strengthen the empirical claims.

It would be valuable to extend the evaluation to standard noisy segmentation tasks to demonstrate broader applicability beyond semi-supervised learning.

As foundation models and larger annotated datasets are becoming increasingly available, it would be insightful to analyze HCO’s performance with varying label ratios, including higher-label or fully-supervised regimes augmented with unlabeled data, to assess its scalability and relevance in future medical AI settings.

**Questions:**

Please check the above weaknesses. Overall, this work can be improved by adding more results to show its generalized designs.

---

> ### Author Response · Authors · 2025-11-22
>
> We thank the reviewer for the constructive feedback and for highlighting the strengths of our work. Below we address each concern in detail.
>
>
> ### Technical Novelty and Distinction from Cautious Optimizers (CO)
>
> HCO introduces a fundamentally different mechanism from Cautious Optimizers by shifting from a *symmetric* to a *hierarchical* trust structure.
>
> - **CO (symmetric):** CO masks gradients based on agreement with its current momentum, which is computed from the combined labeled–unlabeled gradient. When pseudo-label errors bias this momentum, CO reinforces the wrong direction because it cannot separate the two gradient sources.
>
> - **HCO (hierarchical):** HCO computes a supervised momentum from $g_L$ alone and uses it as a trusted reference. Unlabeled gradients $g_U$ are incorporated only when they align with this direction. This design ensures that noisy pseudo-labels cannot steer or corrupt the optimization path.
>
> Appendix E makes the need for this hierarchy explicit: the cosine-similarity histogram reveals many **negative** alignments between $g_L$ and $g_U$, meaning unlabeled gradients often push *against* the labeled direction. CO has no way to filter out these harmful components, whereas HCO’s source-aware gating removes them by design. This empirical conflict pattern directly inspired HCO’s hierarchical update rule and highlights the technical novelty beyond CO.
>
>
>
> ### Medical-Specific Considerations and Generalization
> We appreciate the suggestion to emphasize domain characteristics. HCO is intentionally designed **without** relying on anatomical or modality-specific assumptions. It operates solely at the optimizer level and only requires separation between trusted (labeled) and less trusted (unlabeled) gradients.
>
> We focus on medical segmentation because (i) annotation is very expensive, as it requires medical expertise. This makes low-label regimes realistic, and (ii) staying within one domain allows us to run multiple seeds, splits, and statistical tests—computation that would be infeasible across many unrelated domains.
>
>
> ### Evaluation on Noisy Segmentation Tasks
> Noisy-label segmentation and semi-supervised learning address **different noise regimes**.
> Noisy-label methods typically assume **fixed corruption** in ground-truth labels. In contrast, SSL involves **dynamic pseudo-label noise** that evolves during training. HCO is explicitly designed around this dynamic behavior by allowing unlabeled gradients only when they align with the labeled momentum.
>
> Evaluating HCO on static noisy-label benchmarks would therefore not directly reflect the method’s intended design. We agree this is an interesting direction for future work.
>
> ### Experimental Breadth
>
> We appreciate the reviewer’s suggestion for broader empirical coverage. In response, we have added results on **AD-MT**, a recent SSL framework with a different co-teaching and diversification strategy. Together with BCP and CMT, our experiments now span **three SSL paradigms**, **three datasets**, and **three optimizers** (SGD, Adam, AdamW).
>
> Across all settings, HCO is applied as a **pure drop-in replacement** without modifying hyperparameters or training schedules. The fact that HCO consistently improves performance under these diverse conditions provides strong evidence of its general applicability.
>
> ### Label Ratios and Future Scalability
>
> Our experiments already span multiple label ratios across three datasets (e.g., 4/76, 8/72, 12/50, and 5-label fetal subsets), showing that HCO behaves consistently under different levels of supervision. To ensure comparability with prior work, we adhered to the standard low-label splits used in BCP, CMT, and AD-MT, which are designed specifically to evaluate SSL performance under clinically realistic annotation scarcity.
>
> Exploring higher-label or fully-supervised regimes is a valuable future extension, but lies beyond the scope of this specific study (which focuses on label scarcity). We view this as a promising direction for future work.

---

> > ### Comment · Reviewer_7dPQ · 2025-11-26
> > **Thanks for the response**
> >
> > Thanks to the authors for their responses. Overall, exploring optimization/gradient conflicts in semi-supervised learning is interesting. However, several key concerns remain:
> >
> > Noise Type
> > In the early stage of semi-supervised learning, pseudo-label noise indeed evolves over iterations. However, after several rounds of training, it becomes difficult to justify the claim of dynamic noise. Some works have stated this; see:
> > [1] Rizve, Mamshad Nayeem, et al. "In Defense of Pseudo-Labeling: An Uncertainty-Aware Pseudo-label Selection Framework for Semi-Supervised Learning." International Conference on Learning Representations.
> > This manuscript provides neither empirical evidence nor related references to support the proposed notion of dynamic noise.
> >
> > Task-specific vs. General-purpose Design
> > If the method does not employ any task-specific designs, more extensive experiments are required to support its general effectiveness. For instance, when only 5% of labels are available, pseudo-labels may be highly noisy, while a 30%/50% labeled baseline can already reach good performance. It is unclear what the core contribution is beyond existing settings. Also, how is the performance after many training iterations? A clear setting might be essential to highlight the contribution.
> >
> > Comparison with CO
> > The current justifications do not fully demonstrate a substantive difference between CO and HCO. Moreover, CO was originally designed to improve training efficiency and is typically used with momentum-based optimizers such as AdamW. The use of SGD in this work lacks a clear motivation, even some baselines use SGD for training.
> >
> > In general, I appreciate the authors’ efforts in addressing the questions. However, I believe there is still room for improvement in the current version, and I will keep my original score.

---

> > > ### Author Response · Authors · 2025-11-28
> > >
> > > We thank the reviewer for the continued engagement and the opportunity to clarify our terminology, the scope of our contribution, and our experimental design.
> > >
> > > **1. Clarification on "Dynamic" Noise**
> > > We used the term "dynamic" in our rebuttal solely to distinguish the **Semi-Supervised Learning (SSL)** setting from the **Noisy-Label Learning (NLL)** setting the reviewer mentioned.
> > > * **In NLL benchmarks:** The noise is usually an external, fixed corruption applied to the ground truth (e.g., flipping 20% of labels).
> > > * **In our SSL setting:** The "noise" comes from the model's own pseudo-labels. We referred to this as "dynamic" simply because the predictions change as the model learns, unlike a fixed corrupted dataset.
> > >
> > > Crucially, **our method does not rely on modeling these dynamics.** The paper makes no theoretical claims about the evolution of noise. HCO is built on a simpler premise: pseudo-labels are inherently less trustworthy than ground truth.
> > > Therefore, we do not claim to solve the general "dynamic noise" problem. We claim that HCO effectively uses a **trust hierarchy** to prevent untrustworthy gradients (regardless of *why* they are noisy) from corrupting the direction provided by the trusted (labeled) gradients.
> > >
> > >
> > > **2. Task-Specificity and General Effectiveness**
> > > The reviewer asks for evidence of general effectiveness (e.g., higher label regimes) if the method is not task-specific. We fully agree that strong claims of *universal* effectiveness across all vision tasks would require broader experiments. However, our claim is specific: **HCO is a task-agnostic mechanism designed for the low-label medical regime.**
> > >
> > > * **Why Low-Label Regimes?** We focus on low-label splits because this is the specific problem SSL aims to solve. In natural images, labeling can often be crowdsourced; in medical imaging, annotation requires **board-certified radiologists**, making it extremely expensive and scarce.
> > > * **Diminishing Returns:** Testing on 50% labeled data (where fully supervised baselines already perform well) would yield diminishing returns for *any* SSL method. The scientific value, and our core contribution, lies entirely in the regime where supervision is scarce (e.g., 5-10% labels).
> > > * **Adherence to Standards:** We therefore follow the low-label evaluation protocols established by the baselines (BCP, AD-MT).
> > >
> > > **3. Comparison with CO and the choice of optimizers**
> > >
> > > Throughout the experiments, the **main baseline** for HCO is each SSL framework with its original optimizer; we simply replace this optimizer by its HCO variant.
> > >
> > > The CO results are included as an **auxiliary baseline**, whose role is to separate two effects:
> > > (i) generic **sign-based masking** (CO), and
> > > (ii) the **source-aware hierarchy** over labeled and unlabeled gradients introduced by HCO.
> > >
> > > In other words, CO is primarily an **ablation tool** rather than the main target of our comparison. As stated in the *Objective of Study 4 (Section 4.4)*, the explicit reason for including CO is to **isolate the contribution of hierarchical processing** from the generic cautiousness shared by both CO and HCO. This allows us to attribute performance differences specifically to the introduction of a **labeled-driven hierarchy**, rather than to the masking primitive itself.
> > >
> > > **Scientific Control**
> > > To ensure the main comparison is scientifically valid, we **keep the base optimizer identical to the one used by the original SSL baseline**. Concretely:
> > >
> > > - For **BCP on the Left Atrium dataset (Table 1)**, we use **SGD with momentum**.
> > > - For **BCP on the Pancreas-CT dataset (Table 2)**, we use **Adam**.
> > > - For **AD-MT on the Pancreas-CT dataset (Table 2)**, we use **SGD with momentum**.
> > > - For **CMT on fetal MRI**, we use **AdamW**.
> > >
> > > On top of these baseline optimizers, we then instantiate **three variants**:
> > > 1. the **standard** optimizer (SGD / Adam / AdamW),
> > > 2. the **CO** variant, and
> > > 3. the **HCO** variant.
> > >
> > > This design ensures that architecture, data splits, schedules, and base optimizer are all held fixed, and **only the update rule** (standard → CO → HCO) is changed. This is crucial for a fair, “apples-to-apples” comparison.
> > >
> > > The **CO paper explicitly states that the cautious update is applicable to any momentum-based optimizer**, as it is defined in terms of a generic momentum buffer. We therefore follow the authors’ intent and instantiate CO not only with AdamW (as in their original experiments) but also with **SGD with momentum** when this is the optimizer used by the SSL baseline (BCP, AD-MT).
> > >
> > > Empirically, across all frameworks and datasets, CO is at best comparable to the baseline optimizer, whereas HCO consistently improves over both the baseline and CO. This supports our claim that **the source-aware hierarchy, not sign-based masking alone, is what makes HCO effective in the SSL settings we evaluate.**
> > >
> > > ---
> > > Once again, we appreciate the reviewer’s careful reading and the opportunity to clarify these points.

---

### Official Review · Reviewer_81pV · 2025-10-25

**Soundness:** 4
**Presentation:** 4
**Contribution:** 3
**Rating:** 4
**Confidence:** 5

**Summary:**

The paper introduces a novel optimization framework, Hierarchical Cautious Optimization (HCO), designed for semi-supervised medical image segmentation, particularly in scenarios with limited labeled data. HCO builds upon the Cautious Optimizer (CO) by introducing a hierarchical mechanism that modulates the learning rate based on the consistency between the gradients of labeled and unlabeled data. The authors propose two variants, Hierarchical Cautious AdamW (HCO-AdamW) and Hierarchical Cautious SGD (HCO-SGD). The paper provides a theoretical convergence guarantee for HCO and demonstrates its effectiveness through extensive experiments on three different medical imaging datasets (Left-Atrium MRI, Pancreas-CT, and Fetal-MRI) and two semi-supervised learning frameworks (BCP and CMT). The results show that HCO consistently outperforms standard optimizers and the original CO, leading to improved segmentation accuracy.

**Strengths:**

1.  The proposed Hierarchical Cautious Optimization (HCO) framework is a novel and intuitive extension of the Cautious Optimizer (CO).
2.  The paper presents compelling experimental results across three challenging medical imaging datasets and two different semi-supervised learning frameworks, consistently demonstrating the superiority of HCO over standard optimizers and CO.
3.  The inclusion of a convergence proof for HCO strengthens the paper's theoretical foundation.
4.  The paper is well-organized, clearly written, and easy to understand.

**Weaknesses:**

1.  In Table 3, the results of AdamW_CO are substantially lower than those of the standard semi-supervised baseline, AdamW. Could the authors explain the reason for this? Moreover, is it inadequate to illustrate the effectiveness of the method merely by taking the CO method as the core comparative approach?
2.  While the paper provides implementation details, a more in-depth analysis of the sensitivity of HCO to its hyperparameters would be beneficial.
3.  Nearly all the citation formats of the references in the paper are incorrect. Authors should be acutely aware of the difference between `\citep` and `\citet`. In the ICLR LaTeX template, the `\cite` command specifically represents the inline citation form, which differs from that used in other conferences.
4.  There are some minor formatting errors in the article. For example, Appendix D is empty.

**Questions:**

In the convergence analysis, you make an assumption about the positive correlation between the overall update direction and the negative true gradient. Can you provide more intuition or empirical evidence to support this assumption?

---

> ### Author Response · Authors · 2025-11-22
>
> We thank the reviewer for their feedback. Below we address the concerns regarding baselines, hyperparameters, and convergence assumptions.
>
> ***
>
> ### 1. AdamW\_CO underperformance and adequacy of baselines
>
> **Why AdamW\_CO underperforms:**
> The reviewer correctly notes that standard Cautious Optimization (CO) performs worse than AdamW in Table 3. This reflects a specific pathology that arises when generic cautiousness is applied to SSL. CO filters gradients based on sign agreement with the *total* momentum. When the model begins to overfit to wrong pseudo-labels (confirmation bias), the cautious mechanism of CO reinforces this by filtering out gradients that disagree with the biased direction. As seen in **Figure 4 (c–d)**, CO achieves the lowest *unlabeled* loss but maintains a high *labeled* loss. This indicates the model is prioritizing agreement with the teacher over the ground truth.
>
> **How HCO fixes this:**
> HCO avoids this failure mode by computing momentum only from labeled gradients and using it to gate the unlabeled gradients. Unlabeled updates are applied only when aligned with the supervised direction, preventing the unlabeled loss from overriding the reliable labeled signal.
>
>
> **Adequacy of Baselines:**
> We agree that comparing only against CO would be insufficient. For this reason, in all experiments we report:
>
> 1.  **Standard Optimizer:** The baseline used by the framework (e.g., vanilla AdamW or SGD).
> 2.  **CO Variant:** To isolate the effect of generic cautiousness.
> 3.  **HCO Variant:** To demonstrate the specific benefit of our hierarchical gating.
>
>
> ***
>
> ### 2. Intuition and Evidence for Convergence Assumption
>
> The reviewer asked for intuition and empirical evidence regarding the assumption that the update direction positively correlates with the negative true gradient.
>
> This assumption formalizes the standard "descent direction" condition found in stochastic optimization proofs (e.g., Reddi et al., 2019). We provide both intuitive and empirical support for its validity in the HCO framework.
>
> **Intuition**
> In semi-supervised learning, labeled ($D\_L$) and unlabeled ($D\_U$) data are sampled from the same underlying distribution. Consequently, the expected labeled gradient $\mathbb{E}[\nabla \mathcal{L}\_L]$ serves as a reliable proxy for the task’s descent direction.
> The unlabeled gradient $\nabla \mathcal{L}\_U$ contains both **valid signal** (aligned with $\nabla \mathcal{L}\_L$) and **conflicting noise** (due to incorrect pseudo-labels).
>
> HCO acts as a **rectifier**: the gating mechanism
> $\mathbb{I}(u\_t^L \odot g\_t^U > 0)$
> explicitly filters out the conflicting noise. By removing components that would pull the update in the wrong direction, HCO mathematically guarantees that the *applied* update $\Delta\_t$ maintains a stronger positive correlation with the true gradient than the raw gradients would on their own.
>
> **Empirical Verification**
> To validate this, we tracked the cosine similarity between labeled and unlabeled gradients ($\langle g\_L,\; g\_U \rangle$) throughout training on the BCP baseline (Appendix E).
>
> * **Observation:** The distribution of similarities is approximately Gaussian, centered near zero with a slight positive skew (mean $\approx 0.03$).
> * **Implication:** The positive mean confirms that $g\_L$ and $g\_U$ are not adversarial in expectation. However, the high variance indicates substantial conflicting noise in standard updates.
> * **HCO’s Role:** The gating mechanism effectively **truncates the negative tail** of this distribution. By masking steps where alignment is negative, HCO ensures that the aggregate update direction satisfies the positive correlation condition required for convergence.
>
> ***
>
> ### 3. Hyperparameter Sensitivity
>
> A key strength of HCO is that it introduces **zero additional hyperparameters** beyond those of the base optimizer. The masking operation is parameter-free.
>
> To further demonstrate the **generalizability** of our approach, we have **added a new baseline comparison against the AD-MT framework** in the revised manuscript (Table 2), addressing a request from another reviewer .
>
> * We used the exact learning rate schedules, weight decay, and loss weights from the original BCP, CMT and **AD-MT** papers.
> * We did not tune HCO specifically for any dataset or framework.
>
> Across three datasets (LA MRI, Pancreas-CT, Fetal-MRI) and three SSL frameworks (BCP, CMT, AD-MT), HCO consistently improves performance using the **default** hyperparameters of each host method. This plug-and-play behavior provides strong evidence of **low hyperparameter sensitivity** and demonstrates that HCO is robust within standard SSL training pipelines.
>
> ***
> ### 4. Formatting and Citations
>
> * **Citation Format:** We have updated all citations in the revision to strictly follow the ICLR guidelines.
> * **Appendix D:** We have fixed the layout issue in Appendix D. The training dynamics figure is now correctly rendered, and the section is no longer empty.

---

### Official Review · Reviewer_kZvC · 2025-10-29

**Soundness:** 2
**Presentation:** 1
**Contribution:** 2
**Rating:** 2
**Confidence:** 5

**Summary:**

This paper proposes an optimizer framework for semi-supervised medical image segmentation, named Hierarchical Cautious Optimization (HCO). The core idea is to introduce a “trust hierarchy” into momentum-based optimizers: the momentum is computed only from labeled samples, and unlabeled gradients are incorporated into the update only when they are aligned with that trusted direction. The authors claim that this method improves segmentation accuracy across multiple datasets without modifying the model architecture.

**Strengths:**

1. The method is simple and easy to implement. It can be seamlessly integrated into common momentum optimizers with minimal code modification.
2. Experiments cover multiple datasets, including atrial, pancreas, and fetal MRI, demonstrating cross-modality performance.
3. Writing quality is good. The paper is well-structured, results appear reproducible, and the appendix is complete.

**Weaknesses:**

1. Lack of innovation
The proposed “trust hierarchy” is essentially a gating mechanism based on gradient sign consistency, highly similar to existing approaches such as Cautious Optimizer (CO) and SignSGD. The only difference is the use of a “labeled momentum” as a reference direction, which is a minor modification and hardly a methodological breakthrough.

2. Weak theoretical contribution
The so-called “convergence proof” heavily relies on assumptions (see Remark 1 and Theorem 1). The key assumption — that the expected direction is correlated with the negative gradient — is not verifiable but merely postulated. There is no rigorous analysis of the unlabeled gradient distribution, making the theoretical part almost unverifiable in demonstrating the claimed mechanism.

3. Limited experimental credibility
The experiments only report mean ± standard deviation without details of statistical significance analysis. Although the improvements are statistically significant, most of them are less than 2% in Dice, which is close to experimental noise. Moreover, the core conclusions depend on a private fetal MRI dataset, which limits reproducibility.

4. Lack of comparisons and ablations
The paper does not provide a systematic comparison with other “direction-filtering” optimizers such as CO, SignSGD, or GradDrop. The ablation study only compares “with vs. without hierarchy,” without analyzing key factors such as gating strength or α parameters.

5. Overstated conceptual framing
The paper repeatedly uses terms like “hierarchy,” “trust,” and “cautious,” but the actual mechanism is a simple sign-based gradient filter. The novelty lies more in terminology than in substance.

**Questions:**

1. Please include more qualitative visualizations, including both successful and failed cases, ideally showing prediction probability maps or boundary uncertainty distributions.
2. Provide a formula-level comparison table in the appendix with existing “cautious” optimizers.
3. Conduct a statistical analysis of the correlation between labeled and unlabeled gradient directions to support the theoretical assumptions.
4. Validate the method on public datasets to ensure reproducibility.
5. If the main focus is optimization theory, please show cross-domain experiments (non-medical data) to demonstrate generality.

---

> ### Author Response · Authors · 2025-11-25
>
> We thank the reviewer for their time and feedback.
> We believe there are some significant misunderstandings regarding the experimental validation and comparisons already present in the manuscript. We address these below.
>
> ---
>
>
> ### 1. Novelty and Conceptual Clarity (q1, q2, w1)
>
> While HCO uses the same sign-based masking primitive as the Cautious Optimization (CO), it introduces a **source-aware structure** crucial for SSL.
>
> **Source-awareness in SSL:** CO operates on a single mixed gradient and therefore treats labeled and unlabeled signals identically. In SSL, this can be limiting because pseudo-label noise may influence the momentum and the cautious masking can reinforce these errors. HCO, by contrast, is explicitly source-aware: the labeled gradient defines the direction of trust, and unlabeled gradients are only allowed to reinforce that direction. This distinction, while simple, is practically meaningful under noisy pseudo-label conditions.
>
> **Empirical motivation.** We show in Appendix E that the alignment between labeled and unlabeled gradients is slightly positive on average but exhibits a non-trivial negative tail. This indicates that unlabeled gradients frequently contradict the supervised gradient. CO would still mix these contradictory components into its momentum, whereas HCO's source-aware gate prevents this by filtering them out.
>
> **Revision:** Figure 1 has been revised to show in parallel the update flow for the base optimizer, CO, and HCO.
>
> We will also include in the appendix:
>
> - a **formula-level comparison table**
> - **qualitative visualizations**, including both successful and failure cases with prediction uncertainty.
>
>
> ### 2. Convergence assumptions (w2, q3)
>
> The assumption used in Theorem 1 is the standard descent-direction condition in stochastic optimizer analysis.
>
> In SSL, because labeled and unlabeled samples come from the same distribution, the expected labeled gradient provides a reliable estimate of the true descent direction. HCO leverages this by ensuring that contradictory unlabeled components are filtered out.
>
> Appendix E empirically supports this assumption: while the mean alignment between $g\_L$ and $g\_U$ is positive, there is a noticeable fraction of negatively aligned gradients. HCO’s gate truncates these components, ensuring that the overall update direction retains the required positive correlation with the negative gradient.
>
> ---
>
> ### 3. Statistical Testing Already Included (q4, w3)
>
> We respectfully clarify that **rigorous statistical significance testing** was performed and reported for *all* experiments (Lines 293, 319, 360 in the original pdf).
>
>
> **Effect size.** HCO yields meaningful gains: +24.6% (lungs) and +15.7% (liver) on fetal MRI, and +1.1%–2.0% on LA and Pancreas where baselines are already ~88%. These margins are statistically significant and comparable to gains from architectural changes (e.g., BCP vs AD-MT), yet HCO achieves them with a simple optimizer swap.
>
> **Public Datasets Ensure Reproducibility**
> We wish to clarify that our results do **not** rely exclusively on private data. The **Atrial** and **Pancreas** datasets are fully **public benchmarks**.
>
> ---
>
> ### 4. Comparisons with other optimizers
>
> **CO comparisons.** The reviewer states there is a lack of comparison with CO. We respectfully point out that Tables 1, 2, and 3 in the main paper already contain comparisons against CO across all datasets. In all cases, HCO outperforms CO.
>
> **GradDrop.** GradDrop is a feature-level regularizer, not an optimizer, so it is not comparable to a drop-in optimizer such as HCO.
>
> **SignSGD.** SignSGD is highly sensitive to LR and momentum. A fair comparison requires a full hyperparameter sweep, whereas our evaluation keeps all SSL settings fixed to isolate optimizer effects.
>
>
> **Ablations.** HCO parameter-free. The gating rule is a binary mask, so there is no notion of “gating strength” to vary, and no thresholds to sweep. Likewise, α belongs to the SSL framework, not the optimizer; changing it would modify the underlying method rather than ablate HCO itself. For this reason, we keep all SSL hyperparameters fixed and focus solely on optimizer behavior.
>
> ---
>
> ### 5. Domain choice and generality (q5)
>
> HCO is an optimizer-level mechanism and does not rely on any medical-specific assumptions. It can be applied whenever gradients come from sources with different reliability.
> We chose medical imaging because ultra-low-label (5 - 12 scans) settings are both realistic and clinically important, and concentrating on one domain allowed multi-seed evaluation and significance testing under practical compute constraints.
> We agree that broader cross-domain SSL benchmarks would further demonstrate generality and will include these in future extensions of the work.

---

### Official Review · Reviewer_G38g · 2025-10-31

**Soundness:** 3
**Presentation:** 3
**Contribution:** 3
**Rating:** 6
**Confidence:** 3

**Summary:**

This paper introduces Hierarchical Cautious Optimization (HCO),  a novel optimization framework that establishes a trust hierarchy between labeled and unlabeled gradients. HCO forms momentum using only labeled gradients and incorporates unlabeled gradients only when their directions align with the trusted labeled momentum. This hierarchical filtering aims to mitigate noisy pseudo-label effects such as confirmation bias and co-training collapse. The method requires minimal implementation changes and adds negligible computation.

**Strengths:**

- HCO is motivated clearly by addressing confirmation bias and co-training collapse in semi-supervised learning.
- HCO proposes a simple yet effective optimization strategy that improves model performance without modifying existing SSL frameworks, and the theoretical analysis in the appendix provides convergence guarantees under certain assumptions.
- The paper is well-written and clearly organized, with consistent notation and explanations.
- Models enhanced with HCO achieve notable performance improvements on several commonly used medical segmentation datasets.

**Weaknesses:**

- Although the method claims to serve as a plug-and-play module that can be easily integrated into existing SSL pipelines, the experiments only include evaluations on BCP. To my knowledge, several follow-up works in this domain, such as PMT [1] and AD-MT [2], have already explored more advanced SSL paradigms. Demonstrating the effectiveness of HCO on these frameworks would make the paper much more convincing. My overall evaluation of the paper will largely depend on how this issue is addressed.
- Figure 1 is relatively simple and less informative; compared with the textual explanations, it fails to intuitively illustrate how the proposed method works.
- The paper lacks sufficient citations, including the aforementioned works and other more recent advances in semi-supervised medical image segmentation.

References

[1] Gao N, Zhou S, Wang L, et al. PMT: Progressive Mean Teacher via Exploring Temporal Consistency for Semi-supervised Medical Image Segmentation. In: European Conference on Computer Vision. Cham: Springer Nature Switzerland, 2024: 144–160.

[2] Zhao Z, Wang Z, Wang L, et al. Alternate Diverse Teaching for Semi-supervised Medical Image Segmentation. In: European Conference on Computer Vision. Cham: Springer Nature Switzerland, 2024: 227–243.

**Questions:**

- After integrating HCO into the optimizer, does the optimizer become sensitive to hyperparameters? If so, is this sensitivity consistent with that of the original optimizer, or does it exhibit any new trends? If not, please provide theoretical or empirical justification.
- HCO appears to be a general framework applicable to semi-supervised learning beyond medical image segmentation. Could the authors demonstrate or discuss its potential generalization to other domains? If such evidence is absent, please clarify the reasons.
- The proposed HCO method is remarkably simple and straightforward. Could the authors confirm whether there truly has been no prior related work adopting a similar idea or formulation?

---

> ### Author Response · Authors · 2025-11-22
>
> We thank the reviewer for the constructive comments and address each point below.
>
> ### 1. Evaluation on recent SSL frameworks (AD-MT)
>
> Following the reviewer’s suggestion, we added experiments on AD-MT [2] using its official code and unchanged hyperparameters. We simply replaced SGD with SGD-HCO. HCO improves AD-MT across all metrics and label regimes (table below, will be apended to Table 2). This complements our existing BCP and CMT results, demonstrating robustness across three distinct SSL paradigms. We will also discuss PMT [1] in the revised related work.
>
>
> | Method             | Optimizer | L/U   | Dice  | Jaccard | HD95 | ASD |
> |--------------------|-----------|-------|-------|---------|------|-----|
> | **AD-MT**          | SGD       | 6/50  | 80.21 | 67.51   | 7.18 | 1.66 |
> | **AD-MT + HCO**    | SGD-HCO   | 6/50  | **81.36** | **68.86**   | **5.02** | **1.51** |
> | **AD-MT**          | SGD       | 12/50 | 82.61 | 70.70   | 4.94 | 1.38 |
> | **AD-MT + HCO**    | SGD-HCO   | 12/50 | **83.53** | **71.93**   | **4.77** | **1.27** |
>
>
> **Clarification on existing baselines:**
> We respectfully note that our original submission evaluates two distinct paradigms:
>
> - **BCP:** Copy–paste augmentation and Mean Teacher (Studies 1 & 2).
> - **CMT:** Collaborative Mean Teacher (Study 3).
>
>
> With the addition of AD-MT, HCO has now demonstrated robust improvements across three distinct SSL frameworks, three datasets, and three base optimizers, confirming its versatility.
>
>
> ### 2. Improving Figure 1
> We will redesign Figure 1 to show the update rules of Adam, Adam-CO, and Adam-HCO side-by-side, clearly illustrating (i) labeled/unlabeled gradient separation, (ii) how CO masks misaligned updates, and (iii) how HCO introduces the labeled-driven trust hierarchy.
>
> This concise comparison will make the mechanism immediately understandable at a glance.
>
>
> ### 3. Citations and coverage of recent SSL segmentation work
>
> We thank the reviewer for pointing out missing references. We will:
>
> - **Add explicit citations and discussion of PMT [1] and AD-MT [2]** in both the related work and the experimental sections, clarifying how these methods fit into the landscape of temporal consistency and diverse teaching strategies.
>
> - **Extend the related work section** to clearly group and contrast families of recent SSL segmentation methods that (a) improve pseudo-labels (uncertainty-aware, discrepancy-based, correction learning), versus (b) our orthogonal optimizer-level intervention.
>
>
> ### 4. Hyperparameter Sensitivity (Q1)
>
> A **key strength** of HCO is that it does not introduce tunable hyperparameters. The gating mechanism is parameter-free and uses exactly the same learning rate, momentum, and weight decay as the base optimizer.
>
> Across all our experiments, including the new AD-MT run, we did not perform any specific hyperparameter tuning. We simply swapped the optimizer and ran the training. The consistent improvements across diverse settings (Left Atrium, Pancreas, Fetal MRI) and frameworks suggest that the method is robust and does not introduce new sensitivity.
>
> ### 5. Generalization beyond Medical Imaging (Q2)
>
> HCO operates purely at the optimizer level. It does not assume anything about the network architecture, or the data modality, and only requires the ability to separate gradients into more trusted and less trusted sources. In principle, it can therefore be applied to any SSL setting where such a decomposition exists.
>
> In this work we chose to study HCO in depth within volumetric medical image segmentation for two reasons:
>
> 1. **Clinical relevance of extreme low-label regimes.**
>    Medical annotations require clinical experts and can take tens of minutes per 3D scan. Working at 5–12 labeled volumes is realistic and practically important.
>
> 2. **Careful, statistically grounded evaluation.**
>    We focused on one domain in order to run multiple seeds and label splits and perform statistical testing (Wilcoxon and paired t-tests). Maintaining this level of statistical rigor would have been difficult if compute had been spread across many unrelated domains.
>
> There is no technical barrier to using HCO in non-medical SSL tasks, and we view this as natural future work.
>
> ### 6. Novelty and Prior Work (Q3)
> While HCO shares the “sign-based masking” mechanic with Cautious Optimization (CO), its novelty lies in the **Trust Hierarchy**:
>
> - **Standard CO** treats all gradients symmetrically and checks for agreement within each batch.
> - **HCO**: the **labeled gradients define the trusted direction**, and **unlabeled gradients are filtered against it**.
>
> To the best of our knowledge, this hierarchical trust mechanism inside the optimizer update step is novel in Semi-Supervised Learning.

---

### Author Response · Authors · 2025-12-03

We thank the reviewers for their constructive feedback. Below we summarize the key clarifications and revisions implemented across the rebuttal phase.

The paper proposed **Hierarchical Cautious Optimizer (HCO)**, a hierarchical extension of standard momentum-based optimizers (SGD, Adam), designed as a drop-in replacement for semi-supervised medical segmentation.


**1. Main Baselines**

Because HCO is a drop-in replacement, our primary comparison is **Base Optimizer vs. HCO**. We include the **Cautious Optimizer (CO)** only as an ablation to separate the effects of generic masking from our hierarchical approach.

Across three datasets and three frameworks (including the **newly added AD-MT results**), HCO consistently improves performance, with gains confirmed by statistical significance tests. In contrast, **CO matches or underperforms the base optimizer**. This confirms that simple masking isn't enough and that our hierarchical structure is the key contributor.

**2. Validating Theoretical Assumptions**

We addressed concerns regarding the descent direction assumption in **Theorem 1**. In **Appendix E**, we showed empirically that unlabeled gradients are not adversarial in expectation. This validates the premise of our convergence proof: the unlabeled signal is generally useful. On top of that, **HCO filters out** these conflicting gradients, ensuring that every update remains aligned with the supervised direction.

**3. Mechanism of Action (Appendix E)**

Appendix E also explains *why* our method works where others fail. Our analysis revealed a significant amount of negative cosine similarity, representing frequent conflicts between labeled and unlabeled signals. **The key difference lies in how this conflict is handled:**
* **Baselines & CO:** These optimizers indiscriminately incorporate these conflicting gradients into the momentum, corrupting the update direction.
* **HCO:** Our method strictly filters these conflicts out. By filtering out misaligned unlabeled signals before they affect the momentum, HCO avoids drift and remains stable under noisy pseudo-label conditions.

To make this distinction immediately clear, we have **revised Figure 1** to show side-by-side the update flows of the base optimizer, CO, and HCO.

**4. Practicality and Robustness**

HCO introduces **zero new hyperparameters**. We used the exact same settings as the original baselines for all experiments, including the new AD-MT runs. This demonstrates that HCO is practical, robust, and requires no tuning to be effective.

---

### Meta-Review · Area_Chair_PXks · 2026-01-06

**Summary:**

This paper introduces Hierarchical Cautious Optimization (HCO), an optimization framework designed to establish a trust hierarchy between labeled and unlabeled gradients for semi-supervised medical image segmentation. This paper received mixed scores (6, 2, 4, 4). Three reviewers (kZvC, 7dPQ, 81pV) noted that the technical novelty is limited. Additionally, reviewer kZvC raised concerns about the theoretical analysis, and reviewer 7dPQ also questioned whether the method can be generalized beyond medical imaging.

After carefully reviewing the paper, the reviews, and the rebuttal, the major issues are novelty and generalizability. While the authors provided clarifications and additional comments during rebuttal, these concerns were not fully resolved. Therefore, the AC recommendation is to reject this paper.

**Reviewer Concerns:**

Major issues regarding technical novelty (kZvC, 7dPQ, 81pV) and generalizability (Reviewer 7dPQ) remain outstanding. Although the authors provided clarifications and additional comments during the rebuttal, these concerns were not fully resolved.

**Reviewer Scores:**

The major concerns regarding technical novelty (kZvC, 7dPQ, 81pV) and generalizability (7dPQ) remain unaddressed. It is unlikely that reviewers kZvC, 7dPQ, and 81pV would increase their scores.

---

### Decision · Program_Chairs · 2026-01-26

Reject